# The ubiquitin-proteasome system regulates focal adhesions at the leading edge of migrating cells

**Anjali Teckchandani, Jonathan A Cooper\***

Division of Basic Sciences, Fred Hutchinson Cancer Research Center, Seattle, United States

**Abstract** Cell migration requires the cyclical assembly and disassembly of focal adhesions. Adhesion induces phosphorylation of focal adhesion proteins, including Cas (Crk-associated substrate/p130Cas/BCAR1). However, Cas phosphorylation stimulates adhesion turnover. This raises the question of how adhesion assembly occurs against opposition from phospho-Cas. Here we show that suppressor of cytokine signaling 6 (SOCS6) and Cullin 5, two components of the CRL5$^{SOCS6}$ ubiquitin ligase, inhibit Cas-dependent focal adhesion turnover at the front but not rear of migrating epithelial cells. The front focal adhesions contain phospho-Cas which recruits SOCS6. If SOCS6 cannot access focal adhesions, or if cullins or the proteasome are inhibited, adhesion disassembly is stimulated. This suggests that the localized targeting of phospho-Cas within adhesions by CRL5$^{SOCS6}$ and concurrent cullin and proteasome activity provide a negative feedback loop, ensuring that adhesion assembly predominates over disassembly at the leading edge. By this mechanism, ubiquitination provides a new level of spatio-temporal control over cell migration.

**\*For correspondence:** jcooper@fhcrc.org

## Introduction

During development, wound healing and cancer invasion, migrating cells need to move between other cells and through the dense extracellular matrix (ECM). Cells can attach to and pull on the ECM by using integrins – transmembrane receptors that link ECM outside the cell to focal adhesions (FAs) and the actin cytoskeleton inside the cell (*Alexander et al., 2008*; *Hynes, 2002*; *Pelham and Wang, 1997*; *Petrie et al., 2012*; *Puklin-Faucher and Sheetz, 2009*). FAs are dynamic assemblies containing many proteins held together by dense networks of protein-protein interactions (*Kanchanawong et al., 2010*; *Zaidel-Bar et al., 2007a*). Nascent FAs (often called focal complexes) initiate when talin and other proteins associate with β integrin tails to stabilize an active integrin conformation and stimulate binding to the ECM (*Calderwood et al., 1999*; *Tadokoro et al., 2003*). Talin then binds actin and vinculin and actin flow exerts forces that create additional binding sites for vinculin, which in turn recruits more FA proteins and more actin (*del Rio et al., 2009*; *Jiang et al., 2003*). In this way, the force generated by actin flow, resisted by the ECM, creates a positive feedback loop to stabilize and grow the adhesion (*Case and Waterman, 2015*). In concert, force from the FA acts on actin filaments to induce the formation of contractile stress fibers and actin arcs (*Burridge and Wittchen, 2013*; *Livne and Geiger, 2016*; *Roca-Cusachs et al., 2013*). The contraction of stress fibers and actin arcs provides motive power to advance the cell body. As the cell body moves forwards over an FA, the force vector is redirected and the FA remodels or disassembles, allowing the FA proteins to recycle through the cytosol for reuse at the leading edge (*Wehrle-Haller, 2012*). Inhibition of Rho kinase or myosin relaxes actomyosin tension and induces rapid FA disassembly (*Chrzanowska-Wodnicka and Burridge, 1996*; *Volberg et al., 1994*). These findings

**eLife digest** Animal cells can move in the body, for example to heal a wound, by protruding a leading edge forwards, attaching it to the surroundings and then pulling against these new attachments while disassembling the older ones. Mechanical forces regulate the assembly and disassembly of these attachments, known as focal adhesions, and so do signals from outside the cell that are transmitted to the adhesions via specialized proteins. However, it was not clear how the assembly and disassembly of adhesions is coordinated.

CRL5 is a ubiquitin ligase, an enzyme that can mark other proteins for destruction. Cells migrate more quickly if CRL5 is inhibited, and so Teckchandani and Cooper set out to uncover whether CRL5 affects the assembly and disassembly of focal adhesions. The experiments showed that human cells lacking a crucial component of the CRL5 complex, SOCS6, disassemble adhesions faster than normal cells, but only at their leading edge and not at the rear.

Teckchandani and Cooper also found that SOCS6 localizes to the leading edge by binding to a focal adhesion protein called Cas. Shortly after the attachments assemble, the Cas protein becomes tagged with a phosphate group and then acts to promote the adhesion to disassemble. Further experiments indicated that Cas was marked by the CRL5 complex and possibly destroyed while in or very close to the leading edge adhesions, slowing their disassembly.

Together, these findings suggest that by binding Cas, SOCS6 regulates the turnover of adhesions, specifically by inhibiting disassembly and allowing adhesions to grow at the leading edge. Since SOCS6 is not present in adhesions outside of the leading edge, this may help explain how the older adhesions are disassembled. Future studies could next focus on the exact sequence of events that occur in focal adhesions after the CRL5 complex binds to Cas as the cell migrates.

support a mechanical model in which increased force drives FA assembly and reduced force stimulates disassembly.

FA dynamics are also regulated by signaling through protein kinases and small GTPases. ECM binding stimulates integrin-associated focal adhesion kinase (FAK) and Src-family kinases (SFKs). These tyrosine kinases phosphorylate several FA proteins, helping to build the FA by creating phosphotyrosine (pY) sites that bind to the SH2 domains of additional FA proteins, as well as by activating RhoA (*Parsons et al., 2010*; *Zaidel-Bar et al., 2003*, *2007b*). Thus, FA size increases when endogenous SFKs are activated, consistent with a positive role for SFKs during FA assembly (*Thomas et al., 1995*). However, other evidence suggests that SFKs stimulate FA disassembly. For example, FA size increases and FA turnover is inhibited in fibroblasts with mutations in Src, FAK, or a variety of Src/FAK substrates (*Ilic et al., 1995*; *Ren et al., 2000*; *Volberg et al., 2001*; *Webb et al., 2004*). Furthermore, FAs grow when kinase-inactive Src is over-expressed (*Fincham and Frame, 1998*), and activated mutant Src weakens integrin-cytoskeletal interactions and accelerates FA disassembly (*Felsenfeld et al., 1999*; *Fincham and Frame, 1998*; *Fincham et al., 1995*; *Galbraith et al., 2002*). Therefore, SFKs may regulate both FA assembly and disassembly, perhaps by phosphorylating different substrates.

Cas (p130Cas, BCAR1) is an FA protein that is phosphorylated by Src when integrins engage the ECM (*Fonseca et al., 2004*; *Nojima et al., 1995*; *Petch et al., 1995*; *Vuori and Ruoslahti, 1995*). Cell motility and invasion are stimulated when Cas is over-expressed and inhibited when Cas is deleted (*Brabek et al., 2005*; *Cary et al., 1998*; *Honda et al., 1999*, *1998*; *Huang et al., 2002*; *Patwardhan et al., 2006*; *Sanders and Basson, 2005*). At the molecular level, phosphorylation of Cas by SFKs induces binding to adaptor proteins Crk/CrkL and activation of the Rac1 and Rap1 GTPases (*Hasegawa et al., 1996*; *Sakai et al., 1994*; *Tanaka et al., 1994*; *Vuori et al., 1996*). Forced relocalization of CrkL to FAs stimulates migration via DOCK1 and Rac1 (*Li et al., 2003*). A SFK/Cas pathway stimulates protrusive activity, inhibits stress fibers, and stimulates FA disassembly (*Webb et al., 2004*). Therefore, Cas presents a paradoxical case where phosphorylation is induced by integrin engagement but stimulates FA disassembly and migration.

Interestingly, Cas is mechanosensitive. Cytoskeletal tension induces Cas tyrosine phosphorylation (*Janostiak et al., 2011*; *Tamada et al., 2004*). Mechanical extension of purified Cas promotes its

phosphorylation by SFKs in vitro (*Sawada et al., 2006*). Thus integrin signaling and cytoskeletal tension may increase the level of phosphotyrosine Cas (pYCas) and stimulate FA disassembly whilst simultaneously unfolding talin to promote FA assembly. This raises the question of how FA assembly and disassembly are balanced in such a way that assembly predominates near the leading edge and disassembly occurs under the cell body. Here we provide evidence that CRL5$^{SOCS6}$, a cullin-RING ubiquitin E3 ligase composed of cullin-5 (Cul5), Rbx2, elonginBC and suppressor of cytokine signaling 6 (SOCS6), inhibits pYCas-dependent FA turnover at the front of migrating epithelial cells. We find that SOCS6 is recruited to pYCas in FAs at the leading edge, and that localization of SOCS6 to leading edge FAs slows their disassembly, dependent on cullin and proteasome activity and on binding of SOCS6 to Cas and CRL5. SOCS6 is not present in adhesions under the cell body, and sustained Cas signaling may facilitate their disassembly. The results suggest that CRL5$^{SOCS6}$ targets pYCas within adhesions at the front of the cell, thereby allowing FAs to grow and provide anchorage for stress fibers.

## Results

We previously reported that Cul5 inhibits the epidermal growth factor (EGF)-independent migration of MCF10A epithelial cells (*Teckchandani et al., 2014*). Cul5-depleted cells migrate considerably faster than control cells and have an exaggerated leading lamellipodium bearing tiny FAs, few stress fibers and dynamic membrane ruffles. The increased migration requires Src, suggesting that CRL5 may negatively regulate pY proteins. CRL5 targets pY proteins through the SH2 domains of SOCS substrate receptors. Knock-down experiments suggested that different SOCS proteins regulate different aspects of migration, with SOCS6 targeting pYCas and inhibiting ruffling. We proposed that other SOCS proteins may target additional pY proteins to CRL5 for ubiquitination, and each different target may contribute to a distinct aspect of cell migration (*Teckchandani et al., 2014*).

### Cul5 inhibits adhesion dynamics at the leading edge

To investigate FA dynamics, we wounded monolayers of MCF10A cells that express low levels of EYFP-vinculin and allowed the cells to migrate in the absence of EGF. TIRF microscopy was used to detect EYFP puncta. We previously noted that migrating MCF10A cells contained large FAs and prominent stress fibers along the leading edge, which were lost following Cul5 knockdown (*Teckchandani et al., 2014*). TIRF microscopy of migrating EYFP-vinculin-expressing cells also revealed large FAs at the cell front, but also detected many much smaller FAs all across the ventral surface (*Figure 1a*, left). Large FAs were absent from migrating cells that had been treated with Cul5 siRNA (*Figure 1a*, right, see *Figure 1—figure supplement 1a* for knockdown efficiency). To quantify FA size, we divided the cell into front and back with a line ~ 6 μm from the leading edge (*Figure 1—figure supplement 1b*), and quantified the size distribution of front FAs of control and Cul5-deficient cells. While the number of large FAs at the front of control cells was not sufficient to affect the mean or median FA size, approximately 3.6% of FAs at the front of control cells were larger than 200 pixels (5 μm$^2$), but <0.5% of FAs at the front of Cul5-deficient cells were this large (*Figure 1b*).

To investigate whether different-sized FAs resulted from different dynamics, we recorded time lapse movies. FAs at the front of Cul5-deficient cells were notably more dynamic than those at the front of control cells (*Video 1* and *2*). To illustrate this difference, FAs were color-coded based on their presence in different frames, starting with blue at time zero and advancing through teal, green, orange, red and crimson by 100 min (*Figure 1c*). Many FAs at the front of control cells were white, indicating that they were present during the entire movie, while most FAs at the front of Cul5-deficient cells were singly colored, indicating they were short-lived. Blue FAs, present at the start, marked the original leading edge, and red FAs, born at the end, marked the ending leading edge position. The contrast between stable FAs in control cells and transient FAs in Cul5-deficient cells was also apparent from examination of individual frames from the movies (*Figure 1d*).

To quantify FA dynamics, movies were analyzed using FA analysis software (FAAS) (*Berginski et al., 2011*). This software does not distinguish FAs from the smaller focal complexes, so we refer to all EYFP structures larger than 0.05 μm$^2$ (two pixels) as FAs. We extracted first-order rate constants for the assembly and disassembly phases of each FA (*Figure 1—figure supplement 1c–e*). The rate constants for assembly and disassembly were generally lower for front FAs of control cells

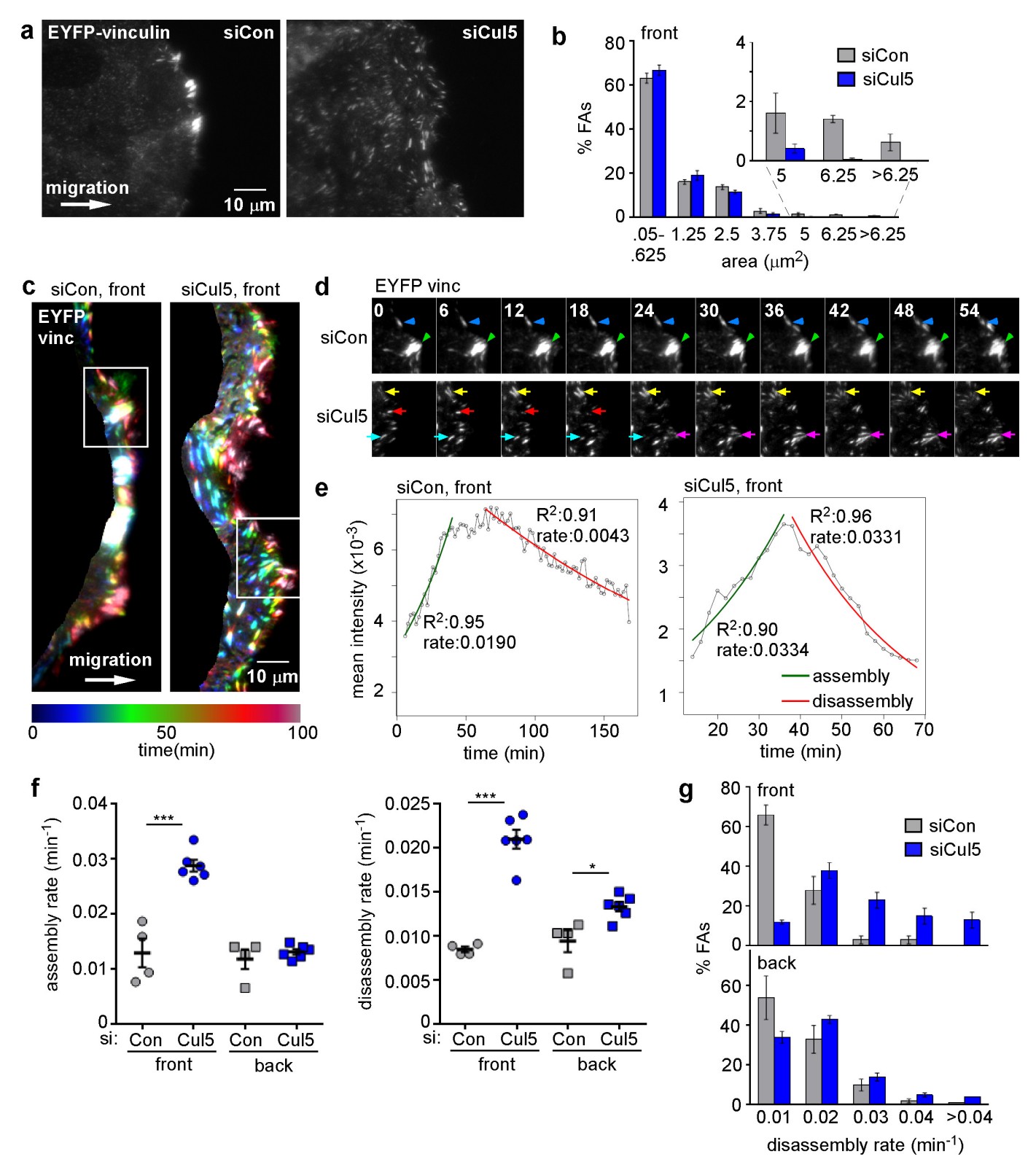

**Figure 1.** Cul5 stabilizes focal adhesions at the front of migrating cells. Focal adhesion dynamics of control and Cul5-deficient MCF10A cells, migrating into a scratch wound in EGF-deficient medium, monitored using EYFP-vinculin. (a) FAs visualized using TIRF microscopy of control and Cul5-deficient cells expressing EYFP-vinculin. (b) Histogram of FA sizes at the front of control (gray) and Cul5-deficient (blue) cells. All structures greater than 0.05 $\mu m^2$ (two pixels) were quantified. The inset shows structures greater than 5 $\mu m^2$ on an expanded scale. (c) Rainbow color representation of FA appearance

*Figure 1 continued on next page*

*Figure 1 continued*

and disappearance. FAs are colored according their presence during the time course, from blue to red. Only the front region of the cell is shown. (**d**) Individual frames from regions boxed in **c**. Arrowheads and arrows indicate stable and dynamic FAs, respectively. (**e**) Automated curve fitting to intensity/time plots for representative FAs from the front of control and Cul5-deficient cells. $R^2$: Pearson's correlation coefficient squared. (**f**) FA assembly and disassembly rate constants from the front (circles) and back (squares) of control (gray) and Cul5-deficient (blue) cells. Mean and standard error of median rates from each of 4–6 time-lapse movies are indicated. *$p<0.05$; ***$p<0.001$. Student's t-test, two tailed, unequal variance. (**f**) Histogram of disassembly rate constants at the front and back of control (gray) and Cul5-deficient (blue) cells. Mean and standard error of biological replicates.

The following figure supplements are available for figure 1:

**Figure supplement 1.** Quantification of FA dynamics.

**Figure supplement 2.** FA size is independent of FA assembly and disassembly rate constants.

than front FAs of Cul5-deficient cells (*Figure 1e*). However, as expected, individual FAs were very heterogeneous. To control for technical or biological variation between experiments, we determined the median assembly and disassembly rate constants of many FAs in each of four to six experiments and then averaged these parameters across independent experiments. We found that FAs at the front and back of control cells assembled and disassembled with similar rate constants (0.012–0.013 min$^{-1}$ for assembly; 0.008–0.009 min$^{-1}$ for disassembly), suggesting that the front FAs were not larger because of increased assembly or decreased disassembly (*Figure 1f*, *Table 1*). Indeed, there was no correlation between mean FA area and assembly or disassembly rate constants (*Figure 1—figure supplement 2*), as expected for first-order reactions. Removing Cul5 had no effect on the assembly or disassembly of back FAs (*Figure 1f*, *Table 1*). At the front, however, removing Cul5 stimulated both assembly and disassembly approximately two-fold, to ~0.029 min$^{-1}$ for assembly

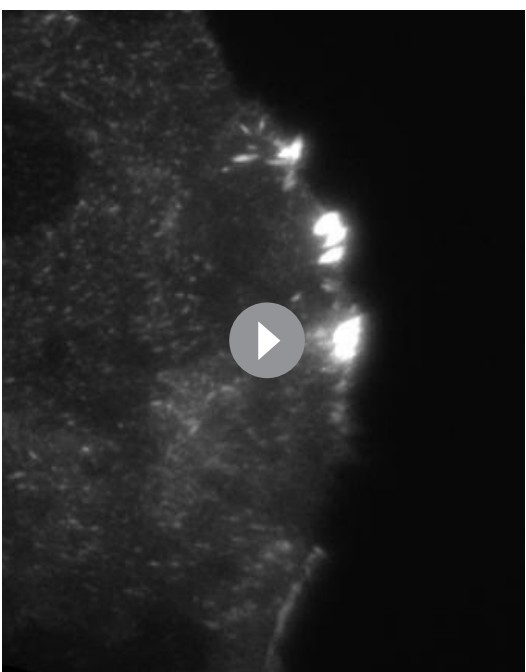

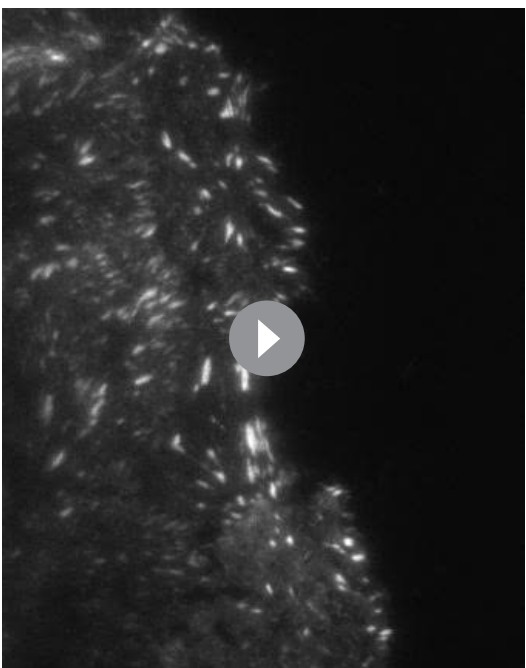

**Video 1.** EYFP-vinculin in migrating MCF10A cells (siCon) imaged every 2 min on a 100× TIRF objective, starting approximately 6 hr after wounding. Length of movie 120 min.

**Video 2.** EYFP-vinculin in migrating MCF10A cells (siCul5) imaged every 2 min on a 100× TIRF objective, starting approximately 6 hr after wounding. Length of movie 120 min.

**Table 1.** Summary of first-order rate constants for FA assembly and disassembly.

| Conditions | Region | Assembly (X $10^{-2}$ $min^{-1}$) | Disassembly (X $10^{-2}$ $min^{-1}$) | # of movies | # of FAs |
|---|---|---|---|---|---|
| siCon | front | 1.3 (±0.2) | 0.8 (±0.03) | 4 | 64 |
| siCul5 | front | 2.9 (±0.1) p=0.0002 | 2.1 (±0.9) p=0.00004 | 6 | 263 |
| siCon | back | 1.2 (±0.2) | 0.9 (±0.1) | 4 | 667 |
| siCul5 | back | 1.3 (±0.05) | 1.3 (±0.5) p=0.04 | 6 | 811 |
| | | | | | |
| siCon | front | 1.1 (±0.1) | 0.8 (±0.1) | 5 | 142 |
| siCul5 | front | 3.0 (±0.4) p=0.008 | 1.9 (±0.08) p=0.00008 | 4 | 444 |
| siCas | front | 1.6 (±0.3) | 0.7 (±0.1) | 6 | 151 |
| siCul5+siCas | front | 1.1 (±0.2) | 0.8 (±0.07) | 5 | 137 |
| | | | | | |
| siCon | back | 0.8 (±0.04) | 0.7 (±0.09) | 5 | 632 |
| siCul5 | back | 0.9 (±0.08) | 0.8 (±0.08) | 4 | 505 |
| siCas | back | 0.8 (±0.05) | 0.6 (±0.08) | 6 | 484 |
| siCul5+siCas | back | 0.7 (±0.1) | 0.5 (±0.05) | 5 | 541 |
| | | | | | |
| shCon | front | 1.0 (±0.1) | 1.0 (±0.1) | 4 | 105 |
| shCul5 | front | 2.0 (±0.1) p=0.0012 | 2.0 (±0.09) p=0.0015 | 4 | 217 |
| shCas | front | 0.9 (±0.1) | 1.0 (±0.06) | 4 | 126 |
| shCul5+shCas | front | 1.0 (±0.2) | 0.8 (±0.07) | 4 | 152 |
| | | | | | |
| shCon | back | 0.9 (±0.06) | 0.7 (±0.8) | 4 | 701 |
| shCul5 | back | 0.8 (±0.04) | 0.9 (±0.1) | 4 | 497 |
| shCas | back | 0.6 (±0.06) | 0.9 (±0.1) | 4 | 559 |
| shCul5+shCas | back | 0.7 (±0.1) | 0.8 (±0.4) | 4 | 608 |
| | | | | | |
| siCon | front | 1.4 (±0.1) | 0.9 (±0.06) | 5 | 185 |
| siSOCS6 | front | 1.7 (±0.2) | 1.4 (±0.03) p=0.0004 | 5 | 201 |
| siCon | back | 0.9 (±0.09) | 0.7 (±0.04) | 5 | 620 |
| siSOCS6 | back | 0.8 (±0.07) | 0.8 (±0.01) | 5 | 469 |
| | | | | | |
| siCon | front | 1.4 (±0.2) | 1.0 (±0.1) | 4 | 254 |
| siSOCS6 (alternate) | front | 1.7 (±0.08) | 1.9 (±0.08) p=0.0007 | 4 | 179 |
| siCon | back | 1.1 (±0.02) | 1.0 (±0.06) | 4 | 640 |
| siSOCS6 (alternate) | back | 1.2 (±0.05) | 1.2 (±0.1) | 4 | 600 |
| | | | | | |
| Con | front | 1.0 (±0.09) | 1.0 (±0.06) | 4 | 76 |
| MLN4924 | front | 2.0 (±0.08) p=0.00016 | 1.6 (±0.03) p=0.0004 | 4 | 111 |
| Con | back | 0.6 (±0.07) | 0.6 (±0.05) | 4 | 182 |
| MLN4924 | back | 1.0 (±0.2) | 0.7 (±0.04) | 4 | 293 |

Rate constants are reported as mean ± SEM

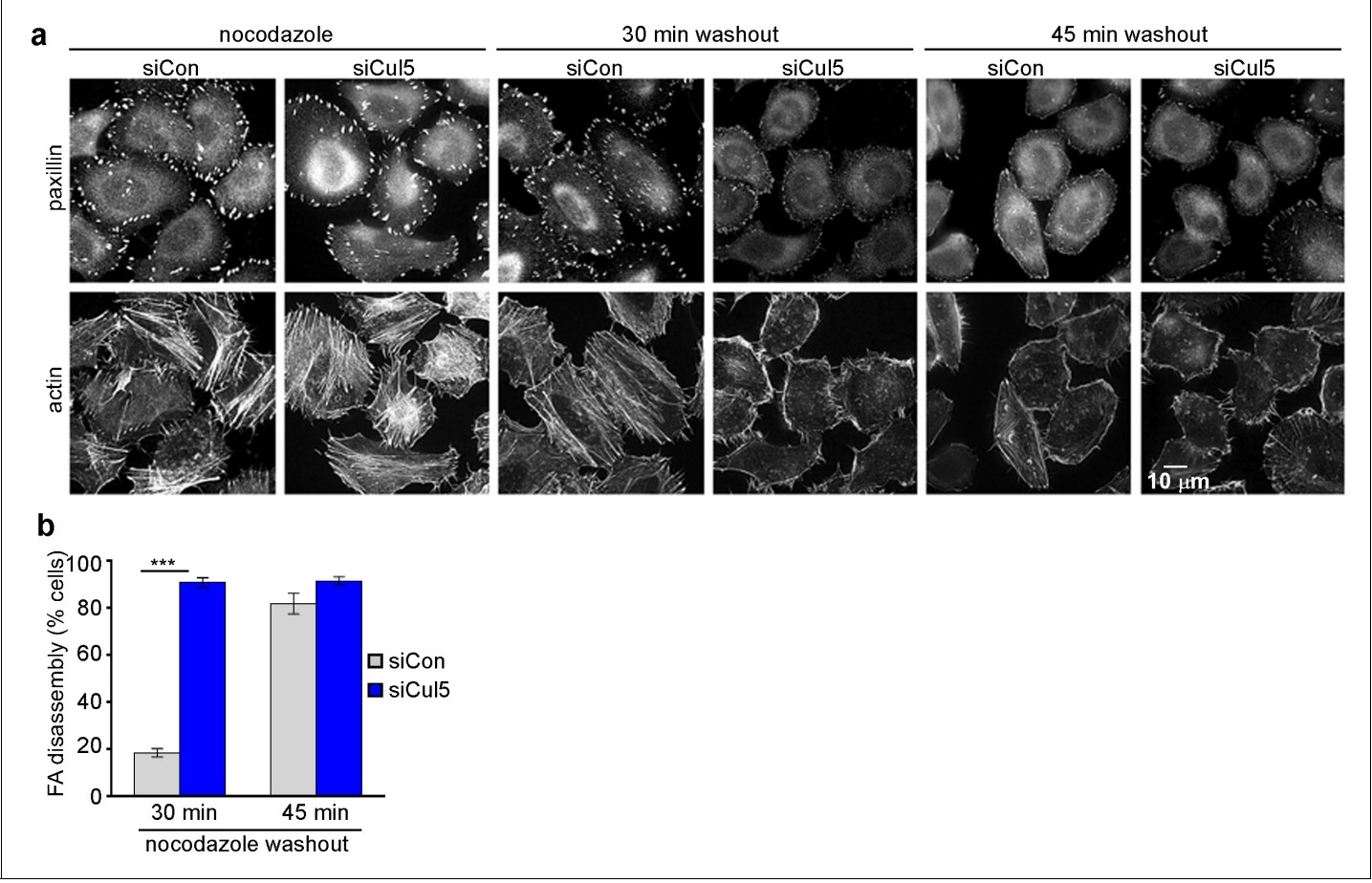

**Figure 2.** Cul5 regulates microtubule-dependent FA disassembly. Control and Cul5-deficient HeLa cells were plated on collagen IV-coated coverslips, serum-starved overnight, and incubated with nocodazole for 3 hr to induce microtubule disassembly and stabilize FAs. Cells were fixed at various times after nocodazole removal and stained for F-actin and paxillin. (b) Cells lacking large FAs and prominent stress fibers were scored as percent of total cells. Mean and standard deviation of three biologically independent experiments. ***p<0.001.

The following figure supplements are available for figure 2:

**Figure supplement 1.** Cul5 does not regulate microtubule growth.

**Figure supplement 2.** Adhesion turnover in Cul5-deficient cells is not regulated by clathrin-mediated endocytosis.

and ~0.021 min$^{-1}$ for disassembly (*Figure 1f*, *Table 1*). Similar assembly and disassembly rates were measured in cells in which Cul5 was stably knocked down with shRNA targeting a different sequence, so the effect is unlikely to be off-target (see below, *Figure 3—figure supplement 1*). The increase in disassembly of leading edge FAs was also apparent from histograms of disassembly rate constants: the percentage of FAs with disassembly rates <0.02 min$^{-1}$ shifted from >60% to <10% when Cul5 was removed (*Figure 1g*). While we were unable to directly measure the lifetimes of the large FAs at the front of control cells, most persisted for longer than the 3 hr duration of the movies, suggesting lifetimes exceeding 3 hr. In contrast, front FAs of Cul5-deficient cells had an average 24 min half time for assembly, 36 min stable phase, and 33 min half time for disassembly, for a total 93 min average lifetime. The longer persistence of FAs at the front of control cells may contribute to their increased size.

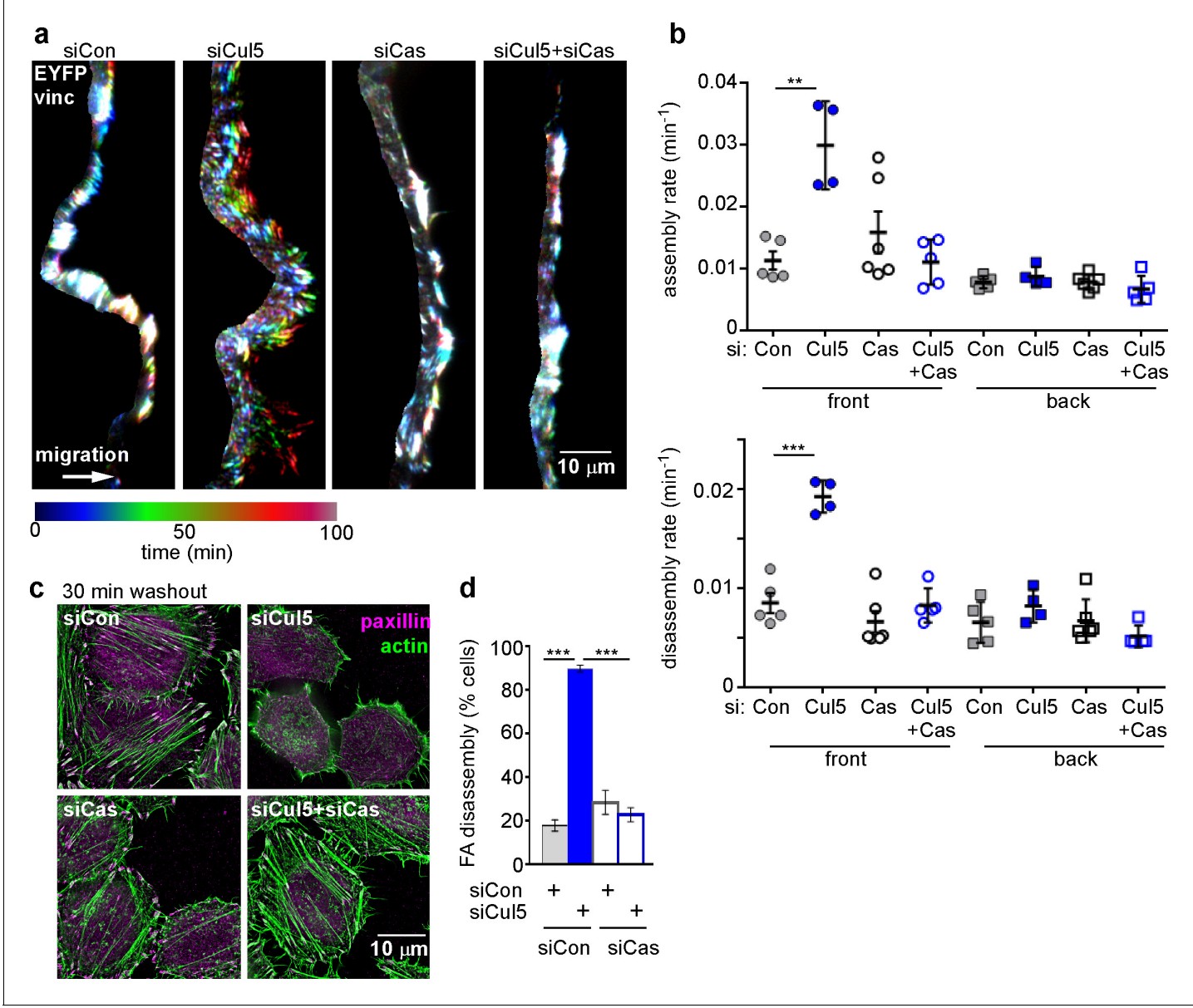

**Figure 3.** Cul5-mediated FA turnover at the leading edge is regulated by Cas. Focal adhesion dynamics of MCF10A cells migrating into a scratch wound in EGF-deficient medium, monitored using EYFP-vinculin and TIRF microscopy. (a) Rainbow color representation of FA appearance and disappearance at the front of control, Cul5-deficient, Cas-deficient, and Cul5-Cas-deficient cells. (b) FA assembly and disassembly rate constants from the front and back. Mean and standard error of median rates from each of 4–6 time-lapse movies are indicated. ***p<0.001. Student's t-test, two tailed, unequal variance. (c,d) Nocodazole washout assay as in *Figure 2*. (c) Images (deconvolution microscopy, ventral section) and (d) quantification of FA disassembly in cells depleted for Cul5 and Cas. Cas was required for the accelerated FA disassembly in Cul5-deficient cells. Mean and standard error of three biologically-independent experiments. ***p<0.001.

The following figure supplement is available for figure 3:

**Figure supplement 1.** Specificity controls for Cul5 and Cas knockdown.

## Cul5 regulates microtubule-dependent FA disassembly

Microtubules (MTs) are required for FA disassembly and are observed repeatedly engaging with FAs during disassembly (*Bershadsky et al., 1996*; *Kaverina et al., 1999*, *1998*; *Stehbens et al., 2012*). We tested whether CRL5-regulated FA disassembly requires MTs. We treated cells with nocodazole

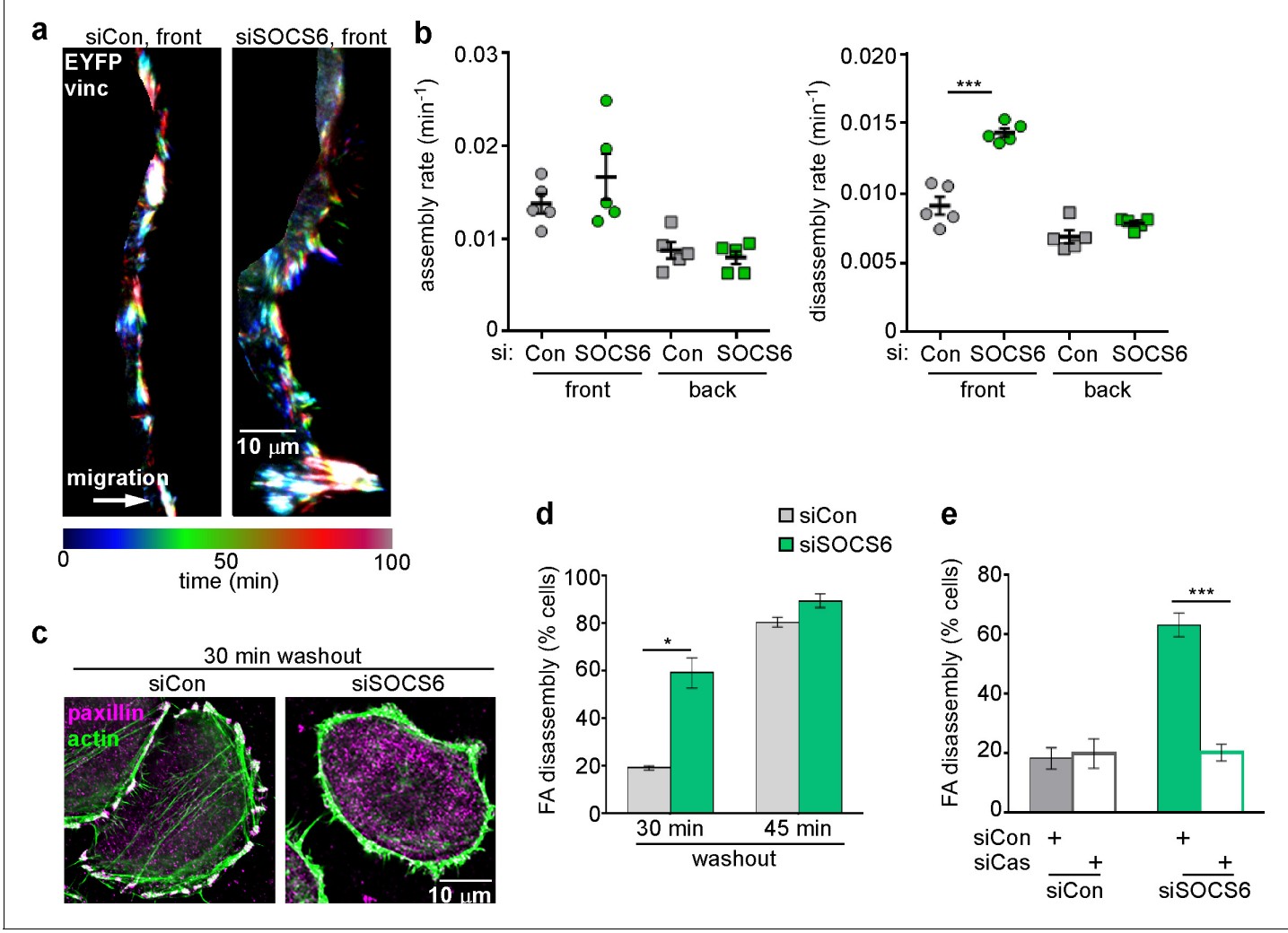

**Figure 4.** FA disassembly in migrating cells is regulated by SOCS6. Focal adhesion dynamics of MCF10A cells migrating into a scratch wound in EGF-deficient medium, monitored using EYFP-vinculin and TIRF microscopy. (**a**) Rainbow color representation of FA appearance and disappearance at the front of control and SOCS6-deficient cells. (**b**) FA assembly and disassembly rate constants from the front (circles) and back (squares) of control (gray) and SOCS6-deficient (green) cells. Mean and standard error of median rates from each of five time-lapse movies are indicated. ***p<0.001. Student's t-test, two tailed, unequal variance. (**c–e**) Nocodazole washout assay as in *Figure 2*. (**c**) Staining for paxillin and F-actin in control and SOCS6-deficient HeLa cells 30 min after nocodazole washout reveals accelerated disassembly in SOCS6-deficient cells. Deconvolution microscopy, ventral section. (**d**) Quantification of FA disassembly in cells depleted for SOCS6. Mean and standard error of three biologically-independent experiments. *p<0.05. (**e**) Quantification of FA disassembly in cells depleted for SOCS6 and Cas. Cas was required for the accelerated FA disassembly in SOCS6-deficient cells. Mean and standard error of three biologically-independent experiments. ***p<0.001.

The following figure supplements are available for figure 4:

**Figure supplement 1.** Specificity of SOCS6 knockdown in MCF10A cells.

**Figure supplement 2.** Regulation of FA disassembly requires SOCS6 interaction with CRL5 and Cas.

to disrupt MTs and then removed nocodazole and assayed adhesion disassembly during MT regrowth (*Ezratty et al., 2005*). We found that nocodazole stabilized FAs in both control and Cul5-deficient cells (*Figure 2a*). However, FAs disappeared more rapidly from Cul5-deficient cells than control cells upon nocodazole removal (*Figure 2a,b*). Stress fibers also disappeared more rapidly from Cul5-deficient cells (*Figure 2*). Accelerated FA disassembly in Cul5-deficient cells was not due to faster MT regrowth after nocodazole removal because the MT cytoskeleton fully recovered in less

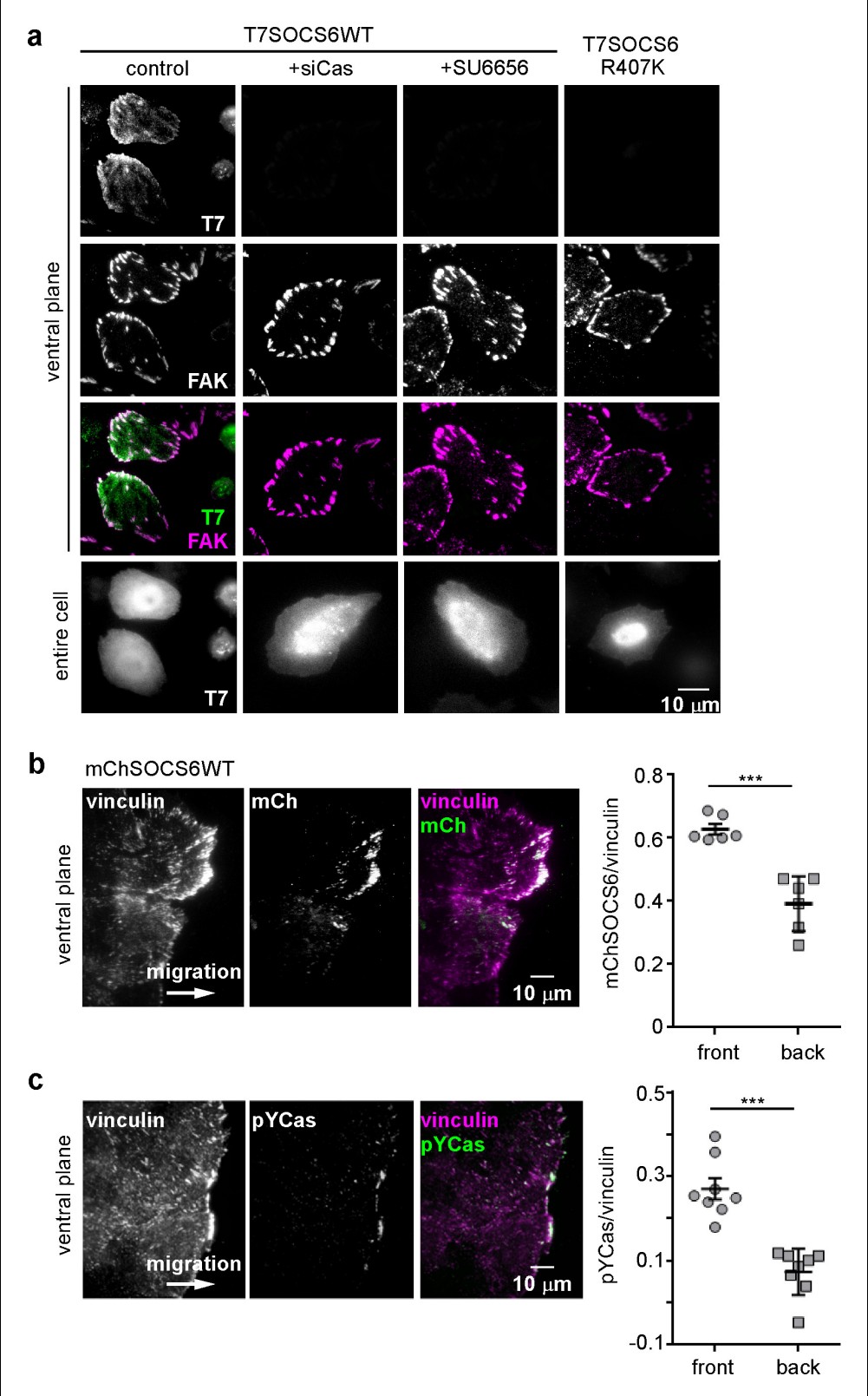

**Figure 5.** SOCS6 localizes to adhesion sites dependent on pYCas. (a) SOCS6 localization to FAs requires Cas, SFK activity, and a functional SH2 domain. HeLa cells were transiently transfected with T7-tagged wildtype (**WT**) or SH2 domain mutant (R407K) SOCS6, treated with or without Cas siRNA, and plated on collagen IV, serum-starved, and incubated with nocodazole for 3 hr to stabilize FAs. One sample was treated with SFK inhibitor SU6656 (10 μM)

*Figure 5 continued on next page*

*Figure 5 continued*

during nocodazole treatment. Fixed cells were stained with antibodies to T7 (green) and FAK (magenta). Images were collected from the ventral plane using TIRF microscopy, and from the entire cell using epifluorescence. (b,c) SOCS6 and pYCas localization in migrating MCF10A cells. (b) MCF10A cells stably expressing mCherry-tagged SOCS6 (mChS6) were allowed to reach confluence, wounded, and allowed to migrate in the absence of EGF. 5 µM MLN4924 was added at the time of wounding and washed off 6 hr later. Cells were fixed 2 hr later and stained with antibodies against vinculin (magenta). Images were collected using TIRF microscopy. The mean ratio of mCherry to vinculin integrated intensity was calculated for 135 FAs at the front and 528 FAs at the back in ~12 cells in two separate experiments. (c) MCF10A cells were allowed to reach confluence, wounded, and allowed to migrate in the absence of EGF. Cells were stained with antibodies against vinculin (magenta) and pYCas (green). Images were collected using TIRF microscopy. The mean ratio of pYCas to vinculin integrated intensity was calculated for 153 FAs at the front and 896 FAs at the back in ~16 cells in two separate experiments.

The following figure supplements are available for figure 5:

**Figure supplement 1.** Focal adhesion localization of SOCS6.

**Figure supplement 2.** Characterization of SOCS6 mutants.

**Figure supplement 3.** Cul5 knockdown increases pYCas in FAs.

---

than 15 min whether or not Cul5 was present (*Figure 2—figure supplement 1a*). Indeed, MT plus ends, tracked with fluorescent plus-end binding-protein EB1, reached FAs in the ventral membrane within minutes of removing nocodazole (*Figure 2—figure supplement 1b*). Therefore, Cul5 slows the disassembly of FAs after MTs have regrown, implying that Cul5 controls the speed of disassembly after MT targeting.

MT targeting is reported to stimulate FA disassembly by clathrin-dependent integrin endocytosis (*Ezratty et al., 2009*; *Ezratty et al., 2005*). Accordingly, depleting clathrin heavy chains (CHC) strongly inhibited FA disassembly in control cells (*Figure 2—figure supplement 2a*). However, CHC depletion only partly inhibited the more rapid FA disassembly in Cul5-deficient cells (*Figure 2—figure supplement 2a*), suggesting that Cul5 inhibits a distinct mechanism of disassembly that is clathrin-independent. Consistent with this interpretation, clathrin-coated pits were missing from the leading lamellipodium of migrating Cul5-deficient cells (*Figure 2—figure supplement 2b*), as recently reported for other cell types (*Kural et al., 2015*). This suggests that Cul5 regulates FA disassembly through mechanisms unrelated to clathrin-mediated endocytosis.

## Cas and SOCS6 regulate FA disassembly at the leading edge

Since we previously identified pYCas as a CRL5 substrate (*Teckchandani et al., 2014*) and since pYCas is required for FA disassembly in migrating fibroblasts (*Webb et al., 2004*), we tested whether Cul5-stimulated disassembly requires Cas. We found that removing Cas suppressed the increase in FA disassembly at the front of migrating Cul5-deficient MCF10A cells but had no effect on FAs at the back or F control cells (*Figure 3a,b*). These effects were unlikely to be due to off-target effects of Cas siRNA because similar results were obtained with cells stably knocked down for Cul5 and Cas using shRNA (*Figure 3—figure supplement 1*). Consistent with a Cul5-Cas pathway, Cas knockdown restored MT-dependent FA disassembly in Cul5-deficient HeLa cells in the nocodazole washout assay, but had no effect on control cells (*Figure 3c,d*). Therefore, Cas is required for increased FA disassembly when Cul5 is absent.

Removal of Cas also suppressed the increased FA assembly of Cul5-deficient cells (*Figure 3a,b*), consistent with the observation that FA and stress fiber assembly is slowed in cas mutant fibroblasts (*Antoku et al., 2008*; *Honda et al., 1998*), potentially due to decreased signaling through a Crk/C3G/Rap1 signaling pathway (*Li et al., 2002*; *Voss et al., 2003*).

SOCS6 is the substrate receptor through which CRL5 binds pYCas (*Teckchandani et al., 2014*). Accordingly, depleting SOCS6 stimulated FA disassembly at the front but not back of migrating MCF10A cells (*Figure 4a,b*). This was unlikely to be due to an off-target effect, because a different pool of SOCS6 siRNA had the same effect (*Figure 4—figure supplement 1*). SOCS6 depletion also

stimulated FA disassembly in the nocodazole washout assay and Cas knockdown inhibited this effect (*Figure 4c–e*). Normal rates of FA disassembly were rescued by re-expression of wildtype SOCS6, but not by mutant SOCS6LCQQ that cannot bind elongin B and the rest of the CRL5 complex (*Figure 4—figure supplement 2a*). Moreover, expression of CasFF, a mutant that does not bind SOCS6 (*Teckchandani et al., 2014*), stimulated FA disassembly regardless of the presence or absence of Cul5, while wildtype Cas only stimulated FA disassembly when Cul5 was absent and Cas15F, a mutant that cannot signal downstream, did not stimulate FA disassembly even when Cul5 was absent (*Figure 4—figure supplement 2b*). These results suggest that SOCS6 inhibits FA disassembly dependent on binding to both Cas and CRL5. However, unlike Cul5, SOCS6 did not impact FA assembly rates (*Figure 4b*), suggesting that, while an increase in pYCas (and other SOCS6 targets) is sufficient to stimulate disassembly, increased pYCas does not accelerate assembly. Other SOCS proteins may target different pY substrates for ubiquitination by CRL5 to regulate FA assembly.

## SOCS6 localizes to FAs by binding to pYCas

CRL5 may regulate FA dynamics by directly interacting with substrates in FAs or by altering the steady-state concentrations of FA proteins in the cytoplasm. We investigated whether CRL5 localizes to FAs. Available antibodies revealed endogenous Cul5 in the nucleus and throughout the cytosol (data not shown). Cul5 was not detectably enriched in FAs. In addition, none of the available SOCS protein antibodies we tested were sufficiently sensitive or specific to detect endogenous SOCS proteins. However, when we transiently transfected HeLa cells with T7 epitope-tagged SOCS6, the tagged protein was enriched in FAs relative to the rest of the ventral membrane (*Figure 5a* and *Figure 5—figure supplement 1*). Enrichment was slight relative to the cytosol, suggesting a low affinity of binding and the potential for rapid exchange.

We investigated the basis for SOCS6 association with FAs. Because FAs contain pYCas and because SOCS6 binds pYCas (*Petch et al., 1995*; *Teckchandani et al., 2014*), we tested whether pYCas is required for SOCS6 localization. FA localization of SOCS6 was inhibited in cells treated with Cas siRNA or with the SFK inhibitor SU6656 (*Figure 5a*). Moreover, a point mutation in the phosphotyrosine-binding SH2 domain of SOCS6 inhibited binding to pYCas (*Figure 5—figure supplement 2a*) and prevented FA localization (*Figure 5a*). Therefore, pYCas is required for SOCS6 localization to FAs.

To test whether SOCS6 also localizes to front FAs in migrating MCF10A cells, we stably expressed wildtype mCherry-tagged SOCS6 (mChSOCS6WT) in MCF10A cells. Unfortunately, stable expression of mChSOCS6WT down-regulated Cas and, perhaps as a consequence, mChSOCS6WT was not detected in FAs (*Figure 5—figure supplement 2b* and data not shown). We attempted to express the mChSOCS6LCQQ mutant, which does not stimulate Cas degradation. However, this mutant was only expressed at low level in MCF10A cells, perhaps due to instability or toxicity of the mutant (*Figure 5—figure supplement 2a,b*). Therefore, we tested whether treating mChSOCS6WT-MCF10A cells with MLN4924, which inhibits the neddylation process required for CRL activity (*Enchev et al., 2015*; *Petroski et al., 2005*; *Soucy et al., 2009*), might allow recovery of Cas levels and localize mChSOCS6WT to FAs. We found empirically that incubation with 5 µM MLN4924 for 6 hr followed by normal media for 2 hr allowed Cas levels to recover without disturbing FA stability (*Figure 5—figure supplement 2c*). Under these conditions, mChSOCS6WT co-localized with vinculin in FAs (*Figure 5b*). Remarkably, mChSOCS6WT was approximately 1.6-fold more abundant in FAs in the front 6 µm than FAs in the rest of the cell. We tested whether leading edge FAs are also enriched for pYCas. The ratio of pYCas to vinculin was approximately 3-fold higher in FAs in the front 6 µm than back (*Fonseca et al., 2004*) (*Figure 5c*). These results suggest that pYCas is enriched in FAs at the front of migrating cells and that SOCS6 preferentially associates with these FAs, as expected if SOCS6 binds to pYCas.

## SOCS6 needs access to FAs to regulate their turnover

Since both Cas and SOCS6 exchange between the cytosol and adhesions, SOCS6 could regulate Cas levels or activity in either location. Inhibition of SOCS6 or Cul5 expression slows Cas turnover, causing a 2–3-fold increase in the steady-state level of Cas (*Teckchandani et al., 2014*) (*Figure 1—figure supplement 1a*). Accordingly, phosphorylated Cas levels in FAs increased approximately 3-fold when Cul5 expression was inhibited, especially in FAs in the front 6 µm of the cell (*Figure 5—*

figure supplement 3). Since Cas exchanges rapidly between the cytosol and FAs (*Janostiak et al., 2011*; *Machiyama et al., 2014*), the increase in pYCas in FAs could be secondary to the increased Cas in the cytosol. Moreover, the chronic increase in pYCas in adhesions could accelerate adhesion turnover regardless of whether SOCS6 targets Cas in the cytosol or in FAs. We therefore tested whether SOCS6 could regulate FA turnover if it was sequestered away from adhesion sites. To this end, we constructed mChS6mito, a fusion of mCherry, SOCS6 and a sequence that targets the mito-chondrial outer membrane (*Bear et al., 2000*). mChS6mito still bound to Cul5 and to Cas (*Figure 6—figure supplement 1a*), but localized to mitochondria instead of FAs (*Figure 6—figure supplement 1b*). In the nocodazole washout assay, mChS6 rescued FA disassembly in SOCS6-depleted cells but mChS6mito did not (*Figure 6—figure supplement 1c*). These results suggested that SOCS6 needs to access FAs in order to inhibit their disassembly.

We used two approaches to test whether CRL5$^{SOCS6}$ represses Cas during FA disassembly. First, we used an optogenetic approach. We used the light-regulated association between the CIBN and CRY2 proteins (*Kennedy et al., 2010*) to sequester SOCS6 away from adhesion sites during the dis-assembly process. We introduced two plasmids into HeLa cells from which endogenous SOCS6 had been removed with siRNA. One plasmid encoded CRY2 fused to mCherry-tagged SOCS6 (CRY2mChS6), and the second encoded CIBN fused to GFP and the mitochondrial targeting sequence (CIBNGFPmito) (*Figure 6a*). In the dark, CRY2mChS6 co-localized with FAK in FAs (*Figure 6b* left). In the light, however, CRY2mChS6 co-localized with CIBNGFPmito at the mitochon-dria and was not detected in FAs (*Figure 6b* right). We then tested whether CRY2mChS6 could res-cue FA disassembly using the nocodazole washout assay. Transfected cells were kept in the dark during growth and nocodazole treatment, so that CRY2mChS6 could diffuse and maintain normal levels of Cas. We then subjected the cells to light or dark conditions and removed nocodazole. The results showed that light inhibits FA disassembly (*Figure 6c*). This implies that SOCS6 must have access to FAs during the disassembly process, and that the steady state level of Cas in the cytoplasm only plays a minor role.

As an independent approach to assess the timing of CRL5$^{SOCS6}$ action, we again used the neddy-lation inhibitor, MLN4924, which inhibits CRL5 and other cullin-RING E3s (*Soucy et al., 2009*). MT-dependent FA disassembly was stimulated when MLN4924 was added an hour before and during 30 min of nocodazole removal (*Figure 6d*). A similar result was obtained with epoxomicin, a protea-some inhibitor (*Meng et al., 1999*) (*Figure 6d*). The results suggest that one or more CRLs and the proteasome need to be active during the disassembly process. Moreover, two hours of treatment with MLN4924 also increased FA assembly and disassembly rates at front but not back of migrating MCF10A cells (*Figure 7a,b*). This short exposure to MLN4924 increased pYCas levels in front but not back FAs and had no effect on total Cas levels (*Figure 7c*). Even though the turnover of many cell proteins is altered by MLN4924 and epoxomicin, the results are consistent with the hypothesis that the activities of Cullins and the proteasome are required during the process of FA disassembly. Combined with the optogenetics results, we suggest that SOCS6 regulates FA turnover locally by direct interaction with pYCas in or near FAs, and not by binding to or altering the steady state con-centration of Cas or other proteins in the cytosol.

The results suggest a model in which FA disassembly is slowed at the cell front by pYCas-depen-dent binding of SOCS6. SOCS6 recruits CRL5 and ubiquitinates pYCas, which is then removed through proteasomal degradation (*Figure 7d*). If SOCS6 or Cul5 is absent, if SOCS6 is held away from FAs, or if cullin or proteasomal inhibitors are present, then pYCas stimulates FA disassembly. A different, clathrin-dependent process prevails under the cell body. SOCS6 traffic thus controls the balance between FA assembly and disassembly in different parts of the cell.

## Discussion

The coordination between FA assembly and disassembly is critical for cell migration. We find that Cul5, SOCS substrate receptors, and the proteasome regulate FA dynamics in epithelial cells. While Cul5 regulates both assembly and disassembly, one of the CRL5 substrate receptors, SOCS6, is required for disassembly but not assembly, implying that other SOCS proteins are involved in FA assembly. SOCS6 inhibits FA disassembly by binding to the FA protein Cas and to the rest of the CRL5 complex. Our results suggest that CRL5$^{SOCS6}$, in conjunction with the proteasome, locally inhibits the SFK-Cas pathway in leading edge FAs to regulate their stability (*Figure 7d*). We

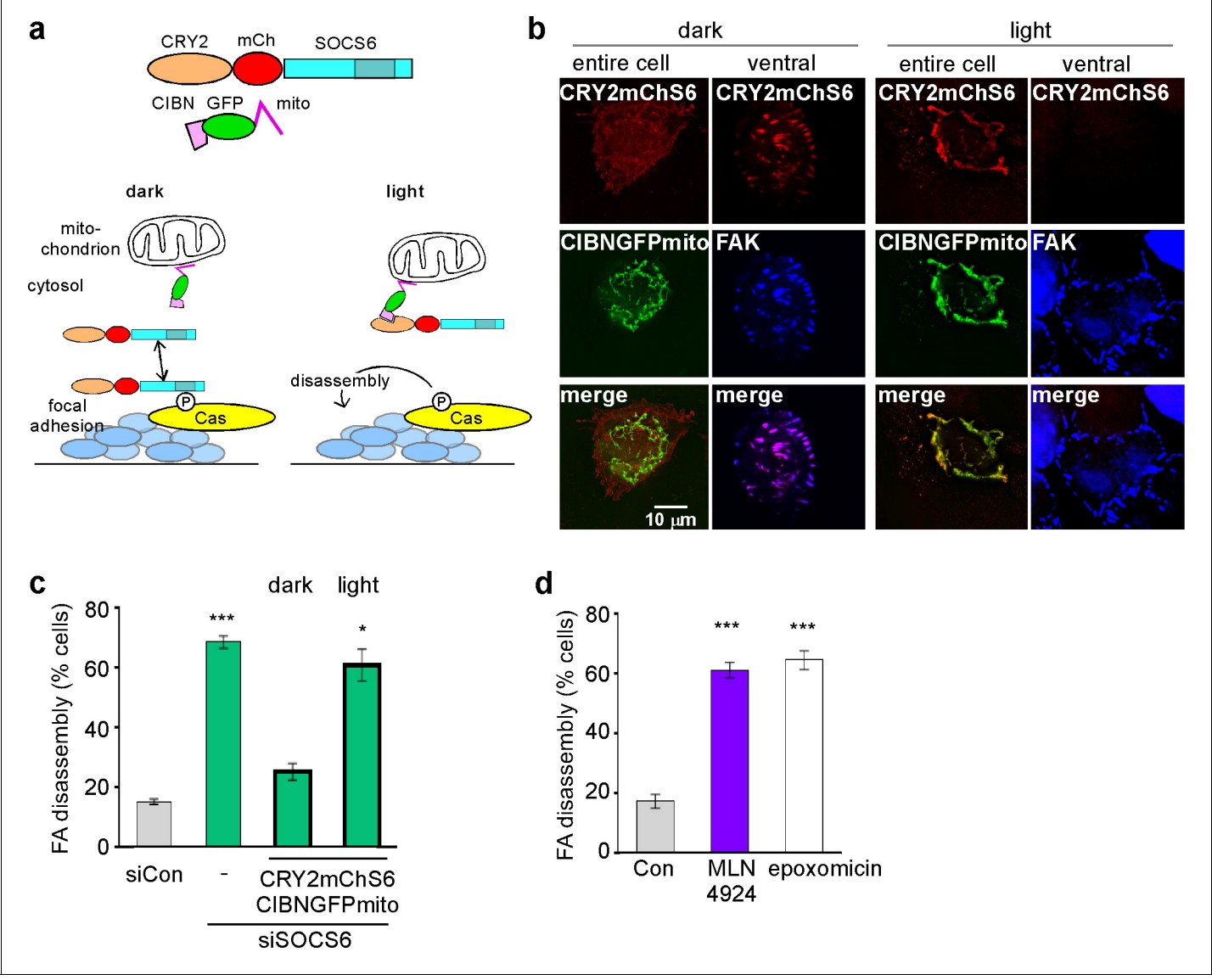

**Figure 6.** SOCS6, Cullin and proteasome activity are required during FA disassembly (**a**) Light-regulated localization of SOCS6. (**b**) Localization of CRY2mChS6 in nocodazole-treated HeLa cells. In the dark, CRY2mChS6 was detected in the cytosol as well as co-localizing with FAK in FAs. However, under blue light illumination, CRY2mChS6 co-localized with CIBNGFPmito at the mitochondria and was not detected in FAs. Entire cell: z-projection, Deconvolution microscopy; ventral section: single plane, TIRF. (**c**) SOCS6 slows FA disassembly only if it has access to FAs during the disassembly process. Cells were plated on collagen IV-coated coverslips, serum-starved overnight, and incubated with nocodazole for 3 hr to induce microtubule disassembly and stabilize FAs. Cells were either kept in the dark during all steps of the assay or illuminated with blue light for 30 min before and during nocodazole washout ('light'). Cells expressing both CRY2mChS6 and CIBNGFPmito were identified by epifluorescence and FA disassembly scored by immunofluorescence for FAK. Mean and standard deviation of three biologically independent experiments. *p<0.05; ***p<0.001. (**d**) Brief treatment with Cullin inhibitor MLN4924 or proteasome inhibitor epoxomicin stimulates FA disassembly. Nocodazole washout assay as in *Figure 2*. Cells were treated with 5 μM MLN4924 or 10 μM epoxomicin 1 hr before washout. Mean and standard deviation of three biologically independent experiments. ***p<0.001

The following figure supplement is available for figure 6:

**Figure supplement 1.** Restricting the access of SOCS6 to FAs speeds FA disassembly.

hypothesize that Cas ubiquitination and degradation occurs close to or in the FA, although we have not shown this directly. Without CRL5$^{SOCS6}$ or proteasome activity, FAs become unstable, stress

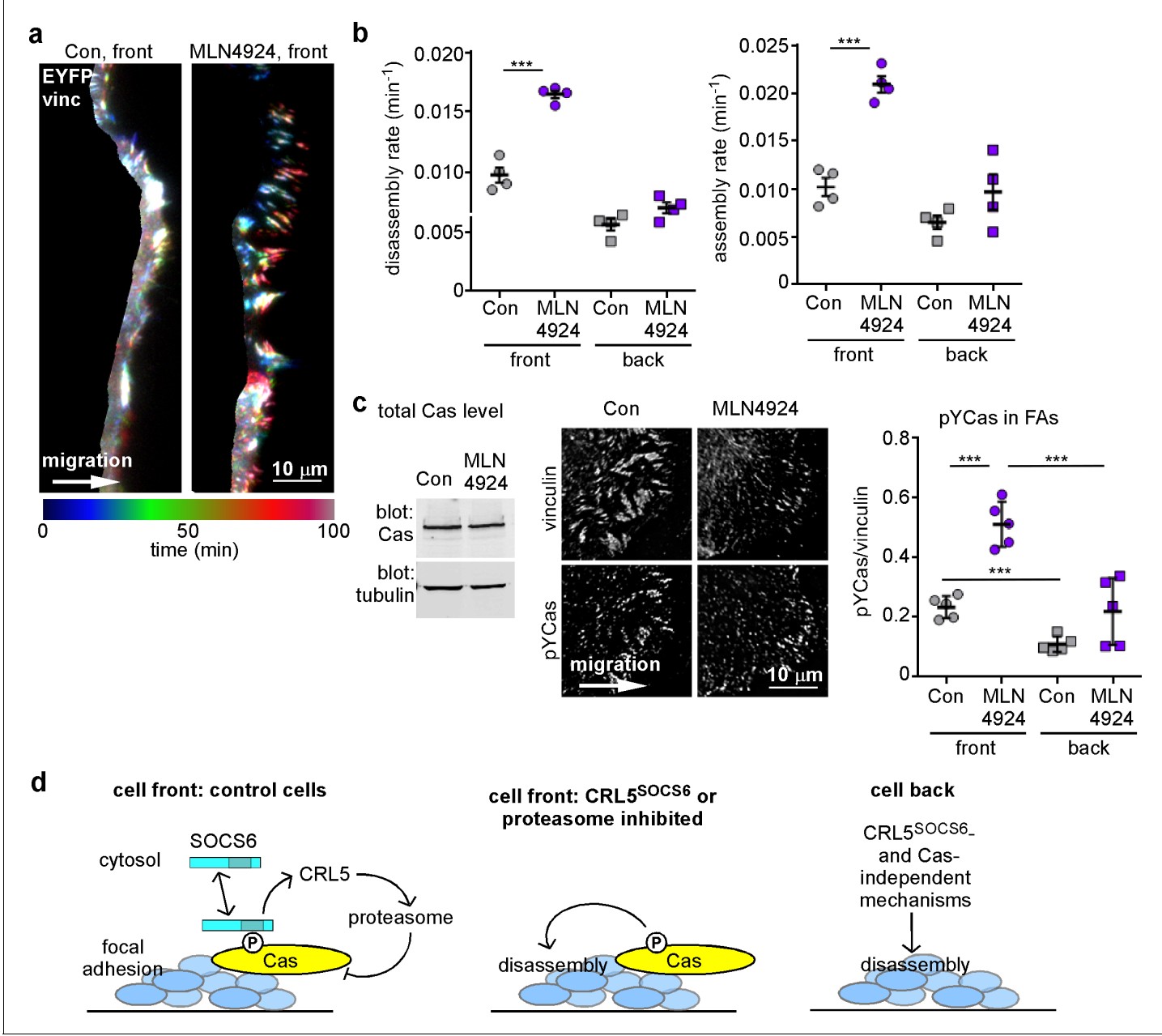

**Figure 7.** FA turnover is inhibited by Cullin and proteasome activity. (**a,b**) Focal adhesion dynamics of control and MLN4924-treated MCF10A cells migrating into a scratch wound in EGF-deficient medium, monitored using EYFP-vinculin and TIRF microscopy. 5 μM MLN4924 was added to cells 2 hr before imaging. (**a**) Rainbow color representation of FA appearance and disappearance at the front of control and MLN4924-treated cells. (**b**) FA assembly and disassembly rate constants from the front (circles) and back (squares) of control (gray) and MLN4924-treated (purple) cells. Mean and standard error of median rates from each of four time-lapse movies are indicated. (**c,d**) MLN4924 increases pYCas locally in FAs at the leading edge but does not increase overall Cas levels. (**c**) Western blot of total cell lysate. 5 μM MLN4924 was added 4 hr before lysis. Cas was not increased on MLN4924 treatment. (**d**) MCF10A cells were allowed to reach confluence, wounded, and allowed to migrate in the absence of EGF. 5 μM MLN4924 was added to cells 4 hr before fixing. Cells were stained with antibodies against vinculin and pYCas. Deconvolution microscopy, ventral section. The mean ratio of pYCas to vinculin intensity (integrated) was calculated for >500 FAs at the front and >1800 FAs at the back in ~16 cells for each condition. (**e**) Summary model. Cas is phosphorylated specifically in FAs at the front of the cell. SOCS6 binds to pYCas in the FAs and locally inhibits pYCas via CRL5 and the proteasome. In cells lacking SOCS6 or Cul5, or in which SOCS6 cannot access FAs or cullins or the proteasome is inhibited, then pYCas stimulates FA disassembly, dependent on microtubules (MTs). In other parts of the cell, slow FA disassembly may occur by MT and clathrin-dependent mechanisms.

fibers are lost and the lamellipodium becomes unstable. Therefore, feedback control through CRL5<sup>SOCS6</sup> organizes FA and actin dynamics at the wound edge and ensures a normal leading lamellipodium. The environment may further regulate SOCS6, CRL5 or proteasome levels or activity, providing additional inputs to modulate the mode and speed of migration. This mechanism may be important in some cancers, where decreased SOCS6 expression has been associated with aggressive disease and increased metastasis (*Fang et al., 2015*; *Letellier et al., 2014*; *Li et al., 2015*; *Wu et al., 2013*; *Zhu et al., 2013*).

In migrating fibroblasts, FA turnover depends on Src and various Src substrates, including FAK, paxillin and Cas (*Fincham et al., 1998*; *Webb et al., 2004*). However, this mechanism has not been reported in other cell types. We found that the SFK-Cas pathway is also required for the rapid FA turnover at the leading edge of epithelial cells in which CRL5<sup>SOCS6</sup>-dependent ubiquitination has been inhibited. This suggests that removing restraints imposed by CRL5<sup>SOCS6</sup> from migrating epithelial cells unveils a latent mechanism more similar to that in migrating fibroblasts, in which Src and Cas, and potentially also FAK and paxillin, stimulate FA turnover.

We and others have observed that Cas phosphorylation is greatest in the large FAs at the front of normal cells and around the edge of spreading cells (*Fonseca et al., 2004*; *Sawada et al., 2006*). Moreover, SOCS6 is specifically recruited to the leading edge adhesions by binding to pYCas, and our optogenetics and inhibitor experiments imply that CRL5<sup>SOCS6</sup> and the proteasome act at the FAs to counteract pYCas and to slow disassembly. Towards the rear, Cas is less highly phosphorylated, there is less SOCS6, and disassembly is slow regardless of SOCS6 or Cul5 activity. This suggests that the regulation of SOCS6 localization is critical for selective stabilization of FAs at the leading edge as cells migrate forwards.

Inhibition of SOCS6-CRL5 did not induce FA disassembly in cells that lack MTs. This suggests that MTs trigger disassembly by a pathway that is inhibited by SFKs and Cas. MT-induced FA disassembly requires the MT motor kinesin-1, presumably to mediate anterograde traffic along MTs towards the FA (*Krylyshkina et al., 2002*). Indeed, MTs deliver a protein complex including IQSEC1, an activator of the Arf6 small GTPase, to FAs (*Yue et al., 2014*). Arf6, together with clathrin, dynein, and clathrin adaptors such as Dab2 and Numb, stimulates integrin β1 endocytosis and MT-dependent adhesion turnover in keratinocytes and HeLa cells (*Chao et al., 2009*; *Ezratty et al., 2009*; *Ezratty et al., 2005*; *Nishimura et al., 2007*; *Yue et al., 2014*). This suggests that MTs deliver endocytic machinery and that integrin endocytosis stimulates FA disassembly. MTs are also involved in weakening the integrin-ECM connection by delivering matrix metalloproteases to FAs (*Stehbens et al., 2014*). However, our evidence suggests that MT-dependent integrin endocytosis is not involved in the SFK-Cas FA disassembly pathway. Clathrin-coated pits are absent from the ventral surface of the leading edge of motile glioblastoma cells (*Kural et al., 2015*) and were absent from the long leading lamellipodium of Cul5-deficient MCF10A cells. Moreover, clathrin knockdown inhibited the slow FA disassembly in normal HeLa cells but not the rapid FA disassembly in Cul5-deficient HeLa cells. Therefore, the MT-dependent signal for FA disassembly in Cul5-deficient cells is unlikely to involve clathrin-dependent endocytosis. The exact nature of the MT trigger for Cas-CRL5<sup>SOCS6</sup> regulated disassembly remains to be determined.

Other studies have implicated ubiquitination in cell migration and FA dynamics. RING protein XRNF185, which destabilizes paxillin via the proteasome, stimulates mesodermal cell migration during convergent extension in Xenopus embryos (*Iioka et al., 2007*). It is not clear whether paxillin ubiquitination occurs locally in this system. On the other hand, paxillin ubiquitination is also stimulated by ubiquitin ligase RNF5, but RNF5 does not induce paxillin degradation and it slows, rather than stimulates, cell migration (*Didier et al., 2003*). Other authors reported that TRIM5 stimulates paxillin turnover and increases FA disassembly. This is the opposite of the effect of CRL5<sup>SOCS6</sup> in inhibiting disassembly (*Uchil et al., 2014*). Ubiquitination may also be involved in an autophagy (ATG) pathway for FA disassembly (*Kenific et al., 2016*). Inhibiting autophagy slows FA assembly and disassembly and increases FA lifetime and size. While it is unclear how autophagy stimulates FA assembly, the evidence suggests that an ATG cargo receptor, NBR1, recruits autophagosomes to FAs to stimulate FA disassembly. Interestingly, NBR1 contains a ubiquitin-binding domain, suggesting that ubiquitination may localize NBR1 to FAs during disassembly.

Perhaps our most surprising finding is that SOCS6 needs to be present in FAs, and cullins and the proteasome need to be active, to regulate FA dynamics. This implies that the localized ubiquitination and degradation of pYCas regulates FA disassembly. Localized degradation of talin by the

protease calpain also triggers FA disassembly (*Bhatt et al., 2002*; *Franco et al., 2004*). However, calpain is an endopeptidase that creates specific cleavage products, and the talin fragments remain in the FA and regulate FA stability (*Huang et al., 2009*). In contrast, ubiquitin-dependent proteolysis typically requires translocation of the ubiquitinated protein into the proteasome (*Ravid et al., 2008*). Removal of Cas by this mechanism would be expected to open up sites in the FA where other Cas molecules could bind. It is known that most of the Cas in a FA exchanges with Cas from the cytosol with a very rapid (~20 s) rate constant (*Janostiak et al., 2011*; *Machiyama et al., 2014*). It seems remarkable that ubiquitination and degradation of pYCas would affect the kinetics of FA disassembly if new Cas molecules could replace degraded molecules with such speed. New approaches will be required to measure the exact sequence of events – Cas binding, phosphorylation, SOCS6 binding, ubiquitination and proteolysis – in FAs during cell migration.

## Materials and methods

### Plasmids

PCR amplification utilized proof-reading Phusion or Herculase enzymes and PCR amplified regions and junctions were sequenced.

pMSCVpuroEYFP-vinculin was made by amplifying murine vinculin with NotI and EcoRI primers from pLenti-H1_CAG_EYFP_C2_mVinculin (*Antoku et al., 2008*) and insertion into pMSCVpuroEYFP-C1.

T7-tagged SOCS and Cul5 constructs have been previously described (*Simo et al., 2013*; *Teckchandani et al., 2014*). LCQQ and R407K mutants were made by Quikchange mutagenesis (Agilent). pCAGHAmCas and pSGTSrcYF have been described previously (*Arnaud et al., 2003*; *Teckchandani et al., 2014*). pLenti CAG EYFPN3 EB1 is an unpublished construct kindly provided by Susumu Antoku. Cas WT, 15F and FF mutants were described previously (*Teckchandani et al., 2014*) and moved into pCAG-T7 by Carissa Pilling.

pmChS6mito was made as follows. Murine SOCS6 was amplified with BamHI and EcoRI primers from pCAGT7SOCS6 (*Simo et al., 2013*) and cloned into Bgl2-EcoRI cut pmChXmito (CAG enhancer/promoter) kindly provided by Russell McConnell and Frank Gertler (*Bear et al., 2000*). The 47 aa mitochondrial targeting sequence is derived from the ActA protein of Listeria monocytogenes. To make pmChS6, pmChS6mito was cut with EcoRI and a self-complementary oligo, AATTGA TGACGTCATC, inserted, abolishing the EcoRI site and adding tandem TGA stop codons.

pBabePuromChSOCS6 was made by amplifying mChXS6 with BamHI and EcoRI primers and insertion into pBabePuro (*Morgenstern et al., 1990*). The LCQQ mutant was made by Quikchange mutagenesis (Agilent).

pCRY2mChS6 and pCIBNGFPmito were made as follows: pCRY2PHR-mCherryN1 and pCIBN(deltaNLS)-pmGFP were obtained from Addgene (*Hughes et al., 2012*). pCRY2PHR-mCherryN1 was cut with NheI and BsrGI and CRY2PHR-mCherryN1 cloned into pmChS6 which had been cut similarly, removing mCh but leaving S6 and the pCAG backbone, to create pCRY2mChS6. pCIBN(deltaNLS)-pmGFP was cut with NheI and AgeI and CIBN(deltaNLS) cloned into pGFPXmito (*Bear et al., 2000*) which had been cut similarly, creating pCIBNGFPmito.

### Cell lines

MCF10A cells were cultured in DMEM/F12 (Thermo Fisher Scientific) supplemented with 5% horse serum (Thermo Fisher Scientific), 20 ng/ml EGF (Thermo Fisher Scientific), 0.5 µg/ml hydrocortisone (Sigma-Aldrich, St. Louis, MO), 0.1 µg/ml cholera toxin (EMD Millipore, Billerica, MA), 10 µg/ml insulin, and penicillin/streptomycin both at 100 U/mL (Thermo Fisher Scientific). MCF10A cells stably expressing Cul5 shRNA, Cas shRNA, pMXpuroII or pLXSH empty vectors have been previously described (*Teckchandani et al., 2014*). MCF10A cells stably expressing EYFP-vinculin from the MSCV promoter were prepared by retrovirus infection with pMSCVpuroEYFP-vinculin and selected with puromycin. MCF10A cells stably knocked down for Cas, Cul5 or both were infected with pLenti-H1_CAG_EYFP_C2_mVinculin and selected by FACS. MCF10A cells stably expressing mCherry-SOCS6 wildtype (WT) or LCQQ were prepared similarly by retrovirus infection with pBabePuromCh-SOCS6 constructs and selected with puromycin. Recombinant retroviruses were packaged using HEK 293 T cells, and infection carried out by standard protocols.

HeLa cells were cultured in DMEM supplemented with 10% FBS and penicillin/streptomycin both at 100 U/ml. Recombinant retroviruses containing pMXpuroII shScrm or shCul5 were packaged using HEK 293 T cells, and infections were carried out by standard protocols. After selection with 2 µg/ml puromycin cells were maintained in the presence of antibiotic.

The identities of the MCF10A and HeLa cells were confirmed by STR DNA profiling. Mycoplasma testing showed that the MCF10A cells were mycoplasma free while the HeLa cells had low level contamination that could be reduced with Primocin (100 µg/ml, two weeks, InvivoGen, San Diego, CA). Primocin-treated Hela cells showed the same Cul5-regulated FA disassembly as untreated cells (compare *Figure 4—figure supplement 2b* and *Figure 2*), suggesting that low level mycoplasma contamination did not affect our results

## Antibodies

The following antibodies were used: mouse anti-paxillin (BD Biosciences, San Jose, CA), rabbit anti-Cas, rabbit anti-FAK, rabbit anti-GAPDH and mouse anti-tubulin (Santa Cruz Biotechnology, Inc., Santa Cruz, CA), mouse anti-vinculin (Sigma-Aldrich), rabbit anti-phospho-p130 Cas (pTyr165) (Cell Signaling Technology, Beverly, MA); mouse anti-T7 (EMD Millipore), rat anti-tyr-tubulin and mouse anti-clathrin (Abcam, Cambridge, MA), mouse anti-HA (HA.11) (BioLegend, Dedham, MA) and rabbit anti-HA (Bethyl, Montgomery, TX). Mouse monoclonal to mCherry was courtesy of Ben Hoffstrom (FHCRC antibody facility) and Jihong Bai-.

## siRNA and DNA transfections

Target sequences and sources of siRNAs and shRNAs are shown in *Table 2*. For siRNA transfection, EYFP vinculin-expressing MCF10A cells were resuspended in growth media and added directly to wells containing 50 pmol pooled siRNA oligonucleotides and Lipofectamine2000 (Thermo Fisher Scientific) on days 1 and 3 for analysis on day 5. Day three transfections were done in uncoated 35 mm fluorodishes (glass thickness 0.17 mm) (World Precision Instruments, Sarasota, FL).

Knockdown experiments in HeLa cells were performed as described (*Teckchandani et al., 2014*). Briefly, cells were transfected with 50 pmol pooled siRNA oligonucleotides using Oligofectamine (Thermo Fisher Scientific) on days 1 and 3, transferred to collagen IV-coated glass coverslips on day four and analyzed on day 5. For rescue and light-activation experiments, DNA was transfected on day 4, cells were transferred to coated coverslips on day five and analyzed approximately 8 hr later.

HeLa cells were grown in 6-well plates to near confluence and transfected with Lipofectamine 2000 (Thermo Fisher Scientific). Cells were transferred to collagen IV-coated coverslips or fluorodishes the next day and imaged 24 hr later.

## Scratch wound assay

MCF10A cells were grown to confluence in uncoated fluorodishes or glass coverslips. Cell monolayers were starved overnight in the assay medium (DMEM/F12 with 2% horse serum, 0.5 µg/ml hydrocortisone, 0.1 µg/ml cholera toxin, 10 µg/ml insulin and 1 ng/ml Mitomycin C (Sigma-Aldrich)), wounded by scratching the surface with a P200 micropipette tip, and the medium replaced with fresh assay medium.

For live cell imaging EYFP vinculin-expressing MCF10A cells were recorded every 2 min for up to 180 min, starting approximately 6 hr after wounding. If required, 5 µM MLN4924 (Active Biochem, Maplewood, NJ) was added 2 hr before recording. A 100×/1.49 CFI Apo TIRF oil immersion objective (pixel size 0.16 µm) on a Nikon Ti fully automated inverted microscope equipped with Perfect Focus, and a stage top incubator with temperature and CO2 control. Images were recorded using an Andor iXon X3 EMCCD camera. Images were acquired using the Nikon NIS Elements software and stacks were assembled using ImageJ. Slight drifts were corrected using the ImageJ registration tool Stackreg. A line was drawn within ~6 µm of the front separating the cell into 'front' and 'back' (*Figure 1—figure supplement 1b*). Rainbow color representations were prepared using Image/Hyperstacks/Temporal-Color Code option in Fiji (http:// fiji.sc).

To detect mChSOCS6 in the front of migrating cells, 5 µM MLN4924 was added at the time of wounding, removed after 6 hr and cells were fixed 2 hr later. To detect pYCas in the front of migrating cells, cells were fixed 6 hr after wounding. Fixed cells were visualized using either TIRF or a 100× NA 1.4 oil objective (pixel size 0.064 µm) on a DeltaVision IX71 microscope (Olympus) equipped

**Table 2.** Target sequences and sources of si and shRNA constructs.

| Reagent | Target | Source |
| --- | --- | --- |
| Cul5 shRNA | 5'-GCTGCAGACTGAATTAGTAG-3' | (*Teckchandani et al., 2014*) |
| Cul5 siRNA pool | 5'-GACACGACGTCTTATATTA-3'<br>5'-CGTCTAATCTGTTAAAGAA-3'<br>5'-GATGATACGGCTTTGCTAA-3'<br>5'-GTTCAACTACGAATACTAA-3' | GE Dharmacon, Lafayette, CO |
| Cas shRNA | 5'-GGTCGACAGTGGTGTGTA-3' | (*Teckchandani et al., 2014*) |
| Cas siRNA pool | 5'-AAGCAGTTTGAACGACTGGA-3'<br>5'-CTGGATGGAGGACTATGACTA-3'<br>5'-CCAGGAATCTGTATATATTTA-3'<br>5'-CAACCTGACCACACTGACCAA-3'* | Qiagen |
| SOCS6 siRNA pool | 5'-CAGCTGCGATATCAACGGTGA-3'<br>5'-TAGAATCGTGAATTGACATAA-3'<br>5'-CGGGTACAAATTGGCATAACA-3'<br>5'-TTGATCTAATTGAGCATTCAA-3' | Qiagen |
| SOCS6 siRNA pool (alternate)[†] | 5'-GAACATGTGCCTGTCGTTA-3'<br>5'-GAAAGTATGCGCTGTCATT-3'<br>5'-TTTAAGCTTGAGCTTTCGCTC-3' | GE Dharmacon<br><br>Thermo Fisher Scientific |
| CHC siRNA pool | 5'-GAAAGAATCTGTAGAGAAATT-3'<br>5'-GCAATGAGCTGTTTGAAGATT-3'<br>5'-TGACAAAGGTGGATAAATTTT-3'<br>5'-GGAAATGGATCTCTTTGAATT-3' | GE Dharmacon |
| siConsh<br>Scrm | 5'-AATTCTCCGAACGTGTCACGT-3'<br>5'-TCGAGCGAGGGCGACTTAACC-3 | Qiagen<br>this paper |

*Also targets mouse Cas. Not used in rescue experiments

[†]Used in *Figure 4—figure supplement 1*.

with an HQ2 CCD camera (Olympus). Images were acquired and deconvolved using SoftWorx (Applied Precision). Deconvolved images from single planes corresponding to the ventral surfaces of the cells or flattened z projections were analyzed using ImageJ (National Institutes of Health).

## Microtubule-dependent focal adhesion disassembly assay

FA disassembly experiments were performed as described by (*Ezratty et al., 2005*) with the following changes. Serum-starved HeLa cells grown at sub-confluent density (~50%) on glass coverslips or fluorodishes coated with collagen IV (2 µg/ml) were treated with 4 µg/ml nocodazole (Sigma-Aldrich) in DMEM with 0.5% BSA and 20 mM HEPES (pH 7.1) for 3 hr to completely depolymerize MTs. After three washes in PBS, cells were left in DMEM with 0.5% BSA and 20 mM HEPES to allow MT regrowth. 10 µM SU6656 (Sugen, San Francisco, CA) was added with Nocodazole. 5 µM MLN4924 and 10 µM epoxomicin (Sigma-Aldrich) were added 1 hr before and during washout.

To score FA disassembly, cells were visualized using a 100× NA 1.4 oil or 60× NA 1.42 oil objective (pixel size 0.11 µm) on a DeltaVision IX71 microscope (Olympus) as described above. FAs below 0.5 µm$^2$ were not distinguishable from noise and were excluded. Cells lacking FAs and prominent stress fibers were scored as percent of total cells. All figures, except *Figure 2—figure supplement 2*, show data from three biologically independent experiments. In every experiment, 20–30 cells were analyzed for each condition.

For live imaging of EYFPEB1 and mChSOCS6 during nocodazole treatment and washout, 100×/1.49 CFI Apo TIRF oil immersion objective, Perfect Focus, and a stage top incubator with temperature and CO$_2$ control were used. Images were acquired using the Nikon NIS Elements software.

## Detection of SOCS proteins in FAs

To detect SOCS proteins in FAs, HeLa cells were transfected with T7-SOCS2, T7-SOCS6 or T7 vector, seeded and serum-starved as described above, treated with 4 µg/ml nocodazole in DMEM with 0.5% FBS and 20 mM HEPES (pH 7.1) for 3 hr, fixed and imaged using a 100×/1.49 CFI Apo TIRF oil

immersion objective in both TIRF and wide-field modes. SOCS6 was localized in migrating MCF10A cells using cells stably expressing pBabePuromChSOCS6WT.

## Light activation experiments

HeLa cells were transfected with siSOCS6 on days 1 and 3 as described above. On day 4, CRY2mChS6 and CIBNGFPmito were transfected using Lipofectamine2000. On day 5, cells were plated on collagen IV-coated fluorodishes. Approximately- 8 hr later they were treated with 4 µg/ml nocodazole for 3 hr in the dark. Washout was either done in the dark or cells were flashed with blue light 30 min before and during nocodazole washout. Blue light illumination was performed using the apparatus described (*Hughes et al., 2012*), with six, 350 mW Royal Blue LEDs positioned 7 cm above a 6-well plate and a duty cycle of 50 ms on, 12 s off. Cells were fixed in the dark under red LED illumination. To confirm that SOCS6 was removed from FAs after blue light illumination, cells were imaged using a 100×/1.49 CFI Apo TIRF oil immersion objective. To score FA disassembly and image SOCS6 in mitochondria cells were visualized using a 100× NA 1.4 oil oil objective on a Delta-Vision IX71 microscope.

## Immunofluorescence

Cells were fixed in formalin at 25°C for 20 min or in methanol for 5 min at −20°C and rehydrated in TBS to visualize microtubules. After permeabilizing with 0.1% Triton X-100 in PBS for 5 min at 25°C, cells were washed in PBS and blocked for 1 hr in 5% normal goat serum/2% BSA in PBS before primary antibody was added for either for 3–4 hr at 25°C or overnight at 4°C. Coverslips were rinsed in PBS before the addition of Alexa Fluor 350-, Alexa Fluor 488-, Alexa Fluor 568- or Alexa Fluor 647-conjugated secondary antibodies, diluted 1:1000 (for deconvolution microscopy) or 1:500 (for TIRF microscopy), for 1 hr at 25°C. Alexa Fluor-tagged phalloidin was used to visualize actin. After several PBS rinses, coverslips were mounted in ProLong Gold solution (for deconvolution microscopy) or left in PBS (for TIRF microscopy).

## Image analysis

To determine FA assembly and disassembly rates, 'front' and 'back' time-lapse movies were uploaded on to the focal adhesion analysis server (FAAS) (*Berginski et al., 2011*) (http:// faas.bme. unc.edu/). The first frame of every movie was used for thresholding. The server returned visualizations showing each frame with every adhesion numbered and outlined (*Figure 1—figure supplement 1c*), as well as images of individual FAs tracked through time (*Figure 1—figure supplement 1d*). These visualizations were used to verify that the adhesions were correctly detected and tracked. To qualify for analysis, an adhesion had to be detected in at least five sequential frames (10 min) and had to be larger than 0.05 $\mu m^2$. Once these parameters were set up, movies were submitted for automated analysis. The server calculated the mean EYFP-vinculin intensity for each FA through time, plotted intensity against time and automatically fitted linear models to the log-transformed time series of intensity values to calculate assembly/disassembly rate constants. FAs with P values less than 0.05 were omitted from the analysis. For each condition, many focal adhesions were measured (*Table 1*). The median rate constants in each of 4–6 experiments were determined and reported.

## qPCR

RNA was extracted and cDNA synthesized. The abundance of Cul5 and SOCS6 RNA was measured by qPCR using QuantiTect SYBR green PCR kit (Qiagen), the 7900HT Real Time PCR System and SDS software (Applied Biosystems). The following primers were used:
Cul5 forward, 5'-TTTTATGCGCCCGATTGTTTTG-3'
Cul5 reverse 5'-TTGCTGGGCCTTTATCATCCC-3'
SOCS6 forward 5'-ATCACGGAGCTATTGTCTGGA-3'
SOCS6 reverse 5'-CTGACTCTCATCCTCGGGGA-3'
GusB forward 5'-AGCGTGGAGCAAGA-3'
GusB reverse 5'-ATACAGATAGGCAG-3'

## Biochemistry

pmChS6mito and control plasmids were co-transfected with pSGTSrcYF and pCAGHACas or empty vector, into HeLa cells with Lipofectamine 2000 (Invitrogen). Cells were stimulated with 2 mM pervanadate for 30 min before lysis in TX100 lysis buffer. Samples were immunoprecipitated with rabbit anti-HA antibodies and protein A beads. Western blots were probed with mouse anti-HA or anti-mCherry.

In parallel, pmChS6mito and control plasmids were co-transfected with pCAGT7Cul5KR or empty vector into HeLa cells with Lipofectamine 2000. Cells were lysed and samples were immunoprecipitated with mouse anti-T7 antibody and protein A beads. Blots were probed with mouse anti-T7 or mouse anti-mCherry.

## Acknowledgements

We are very grateful to Susumu Antoku, Sergi Simo, Russell McConnell, Frank Gertler and Sascha Strait for unpublished constructs and reagents; Zoe Maltzer, Rei Asai, Bruce Huang, and Robyn Emery for technical assistance; Chandra Tucker, Alex Bullock, Rick Horwitz, Michael Yaffe, and our lab members for helpful suggestions; Julio Vasquez of the Fred Hutch Scientific Imaging Resource for assistance with microscopy; the MJ Murdock Charitable Trust for instrumentation funds; Ben Hoffstrom of the Fred Hutch Antibody Resource and Jihong Bai for mCherry antibodies; Mark Berginski and Sean Gomez for establishing and maintaining the FAAS as a resource to the community; and Barry Gumbiner, Julia Maxson, Elizabeth Steenkiste, Youn Na, Carrisa Pilling and Liang Wang for their critical review of the manuscript. This work was supported by grant R01-GM109463 from the US Public Health Service.

## Additional information

### Competing interests

JAC: Reviewing editor, *eLife*. The other author declares that no competing interests exist.

### Funding

| Funder | Grant reference number | Author |
|---|---|---|
| National Institutes of Health | R01 GM109463 | Jonathan A Cooper |

The funders had no role in study design, data collection and interpretation, or the decision to submit the work for publication.

### Author contributions

AT, JAC, Conception and design, Acquisition of data, Analysis and interpretation of data, Drafting or revising the article

### Author ORCIDs

Jonathan A Cooper, http://orcid.org/0000-0002-8626-7827

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
