## [Decision Letter]

Thank you for submitting your article "The ubiquitin-proteasome system regulates focal adhesions at the leading edge of migrating cells" for consideration by *eLife*. Your article has been favorably evaluated by Ivan Dikic as the Senior Editor and three reviewers, one of whom is a member of our Board of Reviewing Editors. The reviewers have opted to remain anonymous.

The reviewers have discussed the reviews with one another and the Reviewing Editor has drafted this decision to help you prepare a revised submission.

Summary:

This is an interesting study that builds upon prior work by these authors that identified a role for CUL5/SOCS6 in negative regulation of cell migration by targeting phosphotyrosine proteins in focal adhesions, including Cas. Here the authors report that CUL5/SOCS6 binding to pY-Cas at the leading edge of cells promotes stability of focal adhesions. The data presented support this conclusion. The authors show that CUL5-SOCS6 functions to inhibit Cas-dependent turnover of focal adhesions at the leading edge but not the rear of migrating epithelial cells. The rates of disassembly are modestly altered ~2-fold with depletion of CUL5. These data were obtained from a number of different cell biology approaches, including image analysis, mitochondrial sequestration, and optogenetic methods. In summary, this study is potentially interesting, but the study could be strengthened by the inclusion of additional control studies and experiments to strengthen the mechanistic understanding of the role of CUL5-SOCS6.

Essential revisions:

1) Many of the studies rely on the use of siRNA pools. Consequently, information about off-target actions of the siRNA is not available. It is important that key observations are replicated with 2 independent single siRNA to confirm the findings presented. Complementation studies using RNAi-resistant mutants would also be helpful.

2) The authors employ MLN4924 which inhibits all cullin-ring E3s. These studies do not establish specificity for the CUL5-SOCS6 RNAi experiments.

3) Figure 1. The selection of 200 pixels as the cut-off seems arbitrary. Is it possible to do image analysis in order to look at the distribution of puncta size across a wide range of pixel distributions?

4) In Figure 3, the authors show that depleting Cas blocks the ability of CUL5 depletion to affect focal adhesion disassembly. However, the mechanism by which Cas is involved is not elucidated in the paper. There are a number of open questions related to mechanism. What is the evidence that CUL5-SOCS6 can ubiquitinate pYCas either in vitro or in vivo? CUL5-SOCS6 can bind pYCas, but what is the evidence that it is pYCas rather than a different protein in the pYCas neighborhood that is the relevant substrate.

5) The authors show that MLN4924 does not stabilize total levels of Cas; consequently, there may only be a small pool of pYCas at the focal adhesions that is ubiquitinated and degraded. Replacement of Cas with a non-Y phosphorylatable Cas mutant might help clarify the mechanism. What is the phenotype of cells overexpressing a SOCS6 protein that does not interact with elongin B?

---

## [Author Response]

*1) Many of the studies rely on the use of siRNA pools. Consequently, information about off-target actions of the siRNA is not available. It is important that key observations are replicated with 2 independent single siRNA to confirm the findings presented. Complementation studies using RNAi-resistant mutants would also be helpful.*

We agree with the reviewers’ concerns about possible off-target effects. We have now repeated experiments using stable knockdown with shRNA or independent pools of siRNA (sequences added to Materials and methods as Table 2) and rescue experiments where possible.

An experiment with shRNAs against Cul5 and Cas is shown in new Figure 3—figure supplement 1. Cas rescue experiments are shown in new Figure 4—figure supplement 2. These results suggest that the effects of Cul5 and Cas knockdown are not due to off-target actions of the siRNA.

We do not have an effective shRNA against SOCS6 at this time. However, the increase in FA disassembly but not assembly that was observed with one pool of siRNAs against SOCS6 was also detected with an independent pool of siRNAs against SOCS6 (new Figure 4—figure supplement 1). Moreover, the previous Figure 6—figure supplement 1, shows that transient transfection of mChSOCS6 rescues normal FA disassembly in the nocodazole washout assay. This rescue has been repeated as a control for transfection of a T7 SOCS6 mutant that does not bind elonginB (new Figure 4—figure supplement 2). For these reasons, we believe that the SOCS6 siRNA effects are unlikely to be off-target. All the new measurements of FA assembly and disassembly rates have been added to Table 1.

*2) The authors employ MLN4924 which inhibits all cullin-ring E3s. These studies do not establish specificity for the CUL5-SOCS6 RNAi experiments.*

We agree that MLN4924 inhibits all cullins and does not establish that Cullin5 is involved. We used MLN4924 as an alternative to the optogenetic approach to assess the time interval during which CRL5^SOCS6^ needs to be active. MLN4924 enters cells and acts rapidly to inhibit Cullin neddylation (within 5 min in HCT116 cells, Supplementary Figure 2 of Soucy et al., 2009). By using MLN4924 we could show that Cullin5 (and perhaps other cullin) activity is only needed during the FA disassembly process, and no change in the steady-state level of Cas is needed (Figure 6 and Figure 7). To avoid confusion regarding the use of MLN4924, we have revised the text as follows:

“As an independent approach to assess the timing of CRL5_SOCS6_ action, we again used the neddylation inhibitor, MLN4924, which inhibits CRL5 and other cullin-RING E3s (Soucy et al., 2009). MT-dependent FA disassembly was stimulated when MLN4924 was added an hour before and during 30 min of nocodazole removal (Figure 6). A similar result was obtained with epoxomicin, a proteasome inhibitor (Meng et al., 1999) (Figure 6). The results suggest that one or more CRLs and the proteasome need to be active during the disassembly process.”

The specificity controls for Cul5 and SOCS6 siRNA have now been added, as described in our response to point 1.

*3) Figure 1. The selection of 200 pixels as the cut-off seems arbitrary. Is it possible to do image analysis in order to look at the distribution of puncta size across a wide range of pixel distributions?*

We have reworded the text associated with the histogram in Figure 1 to clarify the distribution of puncta size. The number of large puncta in the leading edge of control cells is sufficiently small (approximately 4% of the puncta) that the mean and median puncta size is not significantly affected. Instead, we show a histogram, binning puncta by size ranging from 2-25, 26-50, 51- 100, 101-150, 151-200, 201-250, and >250 pixels. (These sizes are converted to μm^2^ in the

figure: 0.05-0.625, 0.625-1.25, 1.25-2.5, 2.5-3.75, 3.75-5.0, 5.0-6.25, and >6.25 μm^2^). This histogram reveals that >60% of puncta are smaller than 25 pixels (0.625 μm^2^) in both cell types, and the size distributions are very similar up to ~200 pixels (5 μm^2^). However, above ~200 pixels there is a clear subpopulation of large puncta in control cells that is absent from Cul5- deficient cells. As the reviewer says, the cut off is arbitrary, but based on what we see. We could calculate a P value for the difference, but the hypothesis is based on the data so we prefer not to.

We assume that the larger puncta are those that attach the stress fibers we saw in fixed cells using phalloidin. As we explain in the paper, we have not distinguished focal complexes from focal adhesions in the analysis. Staining for zyxin, commonly used as a marker for focal adhesions, was uninformative in these cells.

*4) In Figure 3, the authors show that depleting Cas blocks the ability of CUL5 depletion to affect focal adhesion disassembly. However, the mechanism by which Cas is involved is not elucidated in the paper. There are a number of open questions related to mechanism. What is the evidence that CUL5-SOCS6 can ubiquitinate pYCas either in vitro or in vivo? CUL5-SOCS6 can bind pYCas, but what is the evidence that it is pYCas rather than a different protein in the pYCas neighborhood that is the relevant substrate.*

The reviewers raise important points. We can detect Cas ubiquitination when Cas and ubiquitin are over-expressed in HEK293T cells. There is a small decrease in Cas ubiquitination (but not total ubiquitination) when Cul5 is inhibited, suggesting Cul5-dependent and –independent ubiquitination. However, this result does not address whether Cul5-dependent ubiquitination occurs in focal adhesions, which likely only contain a small subset of total Cas molecules. Perhaps, by mapping the ubiquitination site, we will be able to generate site-specific reagents that will allow us to localize Cas ubiquitination in the cell. We therefore prefer to leave this figure for a future paper. Instead, we frankly acknowledge that our model is an interpretation of the data in the modified first paragraph of the Discussion: “[…] Our results suggest that CRL5SOCS6, in conjunction with the proteasome, locally inhibits the SFK-Cas pathway in leading edge FAs to regulate their stability (Figure 7). We hypothesize that Cas ubiquitination and degradation occurs close to or in the FA, although we have not shown this directly.…”.

*5) The authors show that MLN4924 does not stabilize total levels of Cas; consequently, there may only be a small pool of pYCas at the focal adhesions that is ubiquitinated and degraded. Replacement of Cas with a non-Y phosphorylatable Cas mutant might help clarify the mechanism. What is the phenotype of cells overexpressing a SOCS6 protein that does not interact with elongin B?*

We tested whether Cas mutants could rescue microtubule-dependent FA disassembly when Cas was knocked down. As shown in new Figure 4—figure supplement 2, wildtype mouse Cas behaves like endogenous Cas, only activating FA disassembly when Cul5 is absent. A Cas mutant that cannot signal downstream (Cas15F) does not activate FA disassembly whether Cul5 is present or absent. However, the SOCS6-binding mutant of Cas, CasFF, stimulates FA disassembly even when Cul5 is present. This suggests that a direct Cas-SOCS6 interaction is required for CRL5^SOCS6^ to inhibit FA disassembly in migrating cells.

We also tested whether transiently-expressed point mutant SOCS6 that cannot bind ElonginB is able to rescue microtubule-stimulated FA disassembly in SOCS6-deficient HeLa cells. As shown in new Figure 4—figure supplement 1, exogenous wildtype SOCS6 suppresses FA disassembly but the LCQQ mutant does not. These studies confirm that SOCS6-CRL5 interaction is required for suppression of FA disassembly.